# Structural basis for delta cell paracrine regulation in pancreatic islets

Rafael Arrojo e Drigo[1,9], Stefan Jacob [2,10], Concha F. García-Prieto[2,10], Xiaofeng Zheng[1], Masahiro Fukuda [3], Hoa Tran Thi Nhu [4,5], Olga Stelmashenko[1], Flavia Letícia Martins Peçanha[6], Rayner Rodriguez-Diaz[7], Eric Bushong [8], Thomas Deerinck[8], Sebastien Phan[8], Yusuf Ali[1], Ingo Leibiger[2], Minni Chua[1], Thomas Boudier [4,5], Sang-Ho Song [1], Martin Graf[1], George J. Augustine [1], Mark H. Ellisman [8] & Per-Olof Berggren[1,2,7]

Little is known about the role of islet delta cells in regulating blood glucose homeostasis in vivo. Delta cells are important paracrine regulators of beta cell and alpha cell secretory activity, however the structural basis underlying this regulation has yet to be determined. Most delta cells are elongated and have a well-defined cell soma and a filopodia-like structure. Using in vivo optogenetics and high-speed $Ca^{2+}$ imaging, we show that these filopodia are dynamic structures that contain a secretory machinery, enabling the delta cell to reach a large number of beta cells within the islet. This provides for efficient regulation of beta cell activity and is modulated by endogenous IGF-1/VEGF-A signaling. In pre-diabetes, delta cells undergo morphological changes that may be a compensation to maintain paracrine regulation of the beta cell. Our data provides an integrated picture of how delta cells can modulate beta cell activity under physiological conditions.

[1] Lee Kong Chian School of Medicine, Nanyang Technological University, Singapore 636921, Singapore. [2] Karolinska Intitutet, The Rolf Luft Research Center for Diabetes and Endocrinology, Stockholm 171 77, Sweden. [3] Molecular Neurophysiology Laboratory, Signature Program in Neuroscience and Behavioral Disorders, Duke-NUS Medical School, Singapore 169857, Singapore. [4] Bioinformatics Institute (ASTAR) and Image and Pervasive Access Lab (IPAL), Singapore 138632, Singapore. [5] Sorbonne University, UPMC University, Paris 75005, France. [6] Universidade Federal do Rio de Janeiro (UFRJ), Rio de Janeiro 21941-901, Brazil. [7] Miller School of Medicine, University of Miami, Miami 33136, USA. [8] National Center for Microscopy and Imaging Research (NCMIR), University of California San Diego, San Diego 92093, USA. [9] Present address: Salk Institute for Biological Studies, Molecular and Cell Biology Laboratory, San Diego 92037, USA. [10] These authors contributed equally: Stefan Jacob, Concha F. García-Prieto. Correspondence and requests for materials should be addressed to P.-O.B. (email: per-olof.berggren@ki.se)

The islets of Langerhans have a unique cyto-architecture that allows intimate and fine-tuned functional inter-relationships between all structural components[1–3]. Autocrine and paracrine pathways are mediated by islet cells secreting local-acting signalling molecules, such as acetylcholine[4], glutamate[5,6], ATP[7], and GABA[8], in addition to the three peptide hormones insulin, glucagon, and somatostatin. This arrangement forms the basis for an adequate regulation of blood glucose homeostasis under normal conditions, with disturbances in this regulation underlying diabetes pathogenesis[9]. Somatostatin is a powerful paracrine inhibitor of islet alpha and beta cells, which express different isoforms of the somatostatin receptor[10,11]. These receptors enable somatostatin-dependent suppression of glucagon and insulin release[12], indirectly impacting glucose homeostasis[13,14]. Delta cells release somatostatin in response to acetylcholine[15], glutamate[6], urocortin3 (Ucn3)[16], ghrelin[11], and high glucose[17]. Glucose-stimulated somatostatin release depends on action potential firing and subsequent $Ca^{2+}$ influx via L-, T-, R-, and P/Q-type $Ca^{2+}$ channels[17]. In addition, beta and delta cells can regulate glucagon secretion by alpha cells through gap junction communication[18]. However, the molecular machinery underlying delta cell suppression of beta cell or alpha cell activity under in vivo conditions is not clear. We now monitor delta cell structure-function in vivo and in situ using light and electron microscopy, $Ca^{2+}$ imaging and optogenetics in healthy and prediabetic mice. Delta cells within the islet respond to rising blood glucose levels with a synchronized and fast increase in spiking $Ca^{2+}$ dynamics ($[Ca^{2+}]_i$), and this behavior is markedly impaired in prediabetes. Moreover, delta cells have an IGF1/VEGF-A responsive filopodia-like structure that is motile, with spiking $[Ca^{2+}]_i$ behavior and that enables delta cells to contact distant alpha cells and beta cells. Furthermore, in mice with prediabetes, delta cells have impaired $[Ca^{2+}]_i$ dynamics and altered morphology, which may contribute to early stages of beta cell failure and diabetes pathophysiology.

## Results

**Delta cell activity is stimulated by glucose in vivo.** In both rodent and human islets, the somatostatin-secreting delta cells make up 1–5% of the total islet cell population (Fig. 1a). In rodents, delta cells are positioned in the outer islet mantle closer to alpha cells, while in humans they are distributed throughout the islet (Fig. 1a)[1]. Here, we monitored changes in delta cell cytoplasmic free $Ca^{2+}$ concentration ($[Ca^{2+}]_i$) in vivo, as a surrogate of delta cell secretory activity and somatostatin release, using a mouse genetic model expressing the $Ca^{2+}$-binding reporter protein GCaMP3[19] under the control of the SST promoter. Isolated islets isolated from SST-GCaMP3 mice were transplanted into the anterior chamber of the eye (ACE) of C57/BL6 mice[20] and imaged 1 month later with in vivo two-photon microscopy (TPM). Importantly, ACE islets re-vascularize and are re-innervated, making them bona-fide reporters of in situ islet function[20–22]. In fasted animals, delta cells were active with fast $[Ca^{2+}]_i$ spiking activity under resting conditions (resting glucose levels 7–9 mM, Supplementary Movie 1). Next, to investigate the maximum response of delta cells to an increase in circulating glucose levels, we injected insulin (0.25 U/kg, 5–6 min prior to imaging) to induce a mild and transitory state of hypoglycemia (~5.5 mM) before in vivo imaging and glucose stimulation. This did not affect overall glucose homeostasis (Supplementary Fig. 1A). Under basal conditions (i.e., 5.5 mM glucose), ~94% of the cells had detectable $[Ca^{2+}]_i$ spiking activity that increased in response to glucose stimulation (Fig. 1b, Supplementary Fig. 1B–D and Supplementary Movie 2). In these cells, an increase in circulating glucose concentration (from ~5.5 to 18 mM) promoted an increase in $[Ca^{2+}]_i$

spike frequency and amplitude that persisted for up to 4 min after treatment (Fig. 1b–e). At 12 min after stimulation, $[Ca^{2+}]_i$ spikes were markedly reduced in frequency and amplitude as cells entered into a recovery phase (Fig. 1b–d). The remaining ~6% of delta cells exhibited slower, glucose-responsive $[Ca^{2+}]_i$ dynamics (Fig. 1b, Supplementary Fig. 1D). Cells with slow $[Ca^{2+}]_i$ dynamics displayed a significant reduction in $[Ca^{2+}]_i$ frequency and amplitude ~3 min post stimulation, akin to spiking delta cells (Fig. 1b–e). Together, these results indicate that there are two functionally distinct types of delta cells that can be distinguished by their $[Ca^{2+}]_i$ responses to glucose stimulation.

Our results suggest that under basal conditions the activity of individual delta cells within the islet is "uncoordinated" (e.g., each delta cell can fire at different time points) and that glucose stimulation synchronizes delta cell $[Ca^{2+}]_i$ spiking activity (Fig. 1b and Supplementary Movie 1–2). We tested this hypothesis by determining the degree of correlation between delta cells followed by hierarchical cluster analysis (HCA) during basal ($t = 0$–2 min), glucose-stimulated ($t = 2$–6 min), and recovery states ($t > 6$ min). This approach considers the activity profile of each delta cell during the specified time points and determines the degree of correlation (or similarity), which is then ordered by hierarchical clustering. We observed that glucose increases the number of delta cells with similar activity patterns up to 4 min after stimulation, thus creating highly synchronized delta cell clusters within the islet (Fig. 1f). Delta cell synchronization increased $[Ca^{2+}]_i$ spike amplitude in response to glucose (Fig. 1g). While the factors that synchronize delta cells remain to be determined, it is possible that factors secreted from neighboring cells, the autonomic nervous system and/or gap junction communication may contribute to delta cell synchronization[11,15,18]. Our findings support the idea that somatostatin and insulin are present in the islet interstitium at the same time, as suggested by previous in vitro studies[23]. However, different delta cell $[Ca^{2+}]_i$ dynamics have been observed in isolated islets in vitro[11,24], which underscores the likely contributions of other regulatory pathways that are present in the in vivo setting, such as the vasculature and the autonomic nervous system[25].

**Delta cells reach out towards other endocrine cells.** Delta cells are elongated, typically having a well-defined cell soma and a filopodia-like extension[26,27]. We hypothesized that the filopodia-like structures compensate for the low density of delta cells (Fig. 1), by enhancing the range of somatostatin action within the islet and used confocal and electron microscopy to visualize these structures in rodent and human delta cells (Fig. 2a–c, Supplementary Fig. 2A). First, we measured filopodia length in rodent and human pancreas sections (Fig. 2d). The length varied from 1 to 18 μm, which is in agreement with previous reports[26]. Next, we studied the dynamics of delta cell filopodia in vivo using TPM imaging of SST-ChR2-YFP mouse islets transplanted to the ACE. This animal model was chosen because (i) ChR2 channels are tagged with a fluorescent protein (YFP) and thus enable efficient visualization of the cell membrane in vivo and in vitro and (ii) it allows optogenetic stimulation of delta cells (see below). In addition, we used TPM imaging instead of conventional confocal microscopy to prevent significant ChR2 activation[28]. Delta cell filopodia were significantly longer in vivo than in fixed sections, ranging from 2 to 27 μm (Fig. 2d, and a 3D rendering of an in situ Sst-ChR2 delta cell is shown in Supplementary Movie 3). This difference is likely caused by shrinkage associated with paraformaldehyde fixation. In both mouse and primate islets in situ, somatostatin-positive filopodia could be observed in close contact with alpha, beta, and other delta cells that were several microns away (Supplementary Fig. 2B–D).

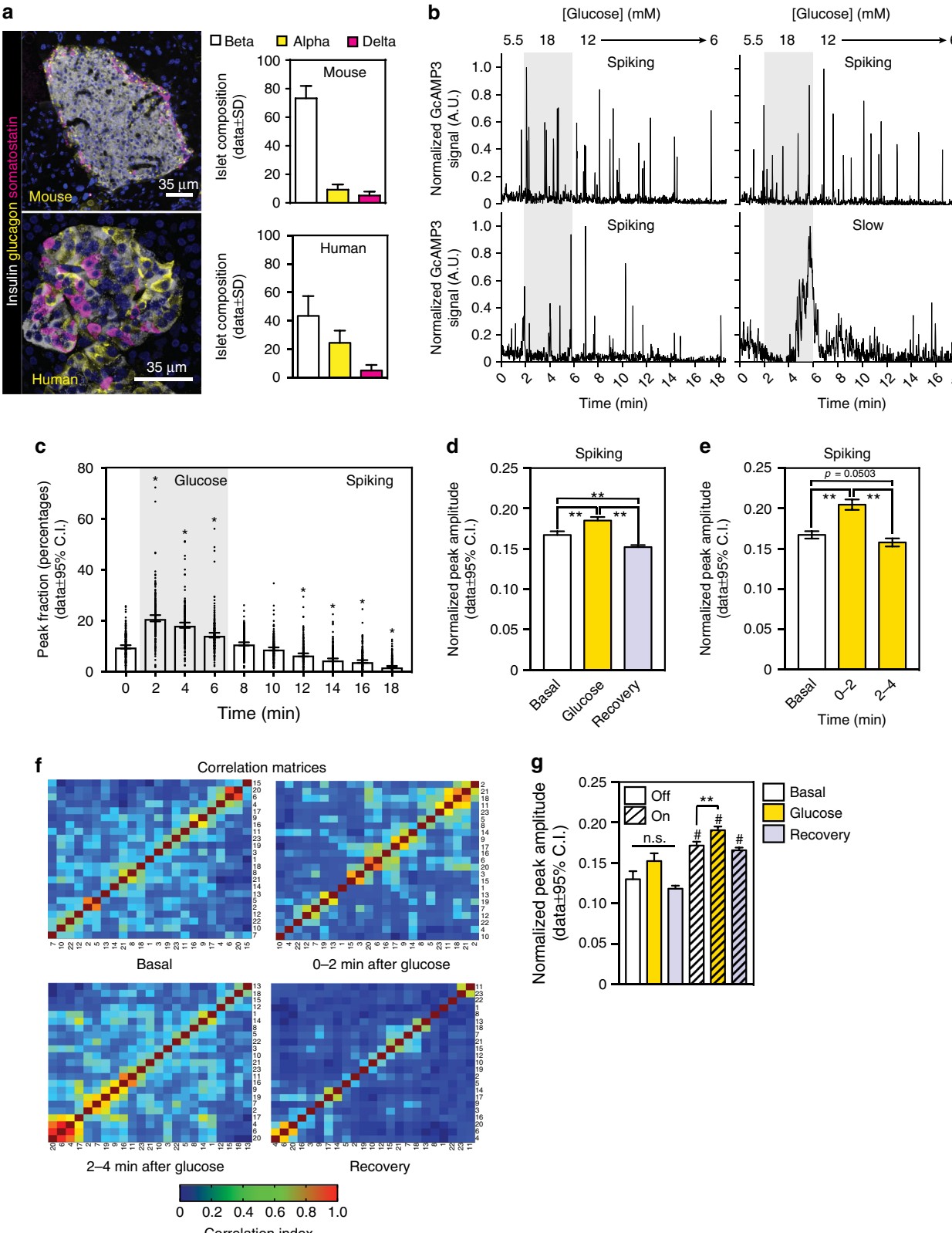

These results suggest that delta cells could contact islet cells beyond their immediate vicinity. To test this hypothesis, we used automated cell segmentation algorithms[29] to identify delta cells and determine the physical advantage provided by filopodia within the crowded islet environment (Supplementary Fig. 3A–G). Using measurements collected in vivo (2–27 µm, Fig. 2d), we observed that increasing the length of delta cell filopodia significantly increased the number of contacts with beta and alpha cells within an islet (Fig. 2e, f, Supplementary Fig. 3A–J). While this effect was modest in rodents, in human

**Fig. 1** Delta cell activity in vivo. Delta cells are activated by increasing glucose levels. Longitudinal imaging of in vivo activity in islets expressing the $Ca^{2+}$ reporter GCaMP3 in delta cells. **a** Maximum projection images a representative mouse (top) and a human islet (bottom) immunostained with anti-glucagon, anti-insulin, and anti-somatostatin antibodies. Cell nuclei are shown in light blue. Graphs on the right show the cellular composition of mouse ($n = 24$ islets total, $n = 7$ mice) and human islets ($n = 21$ islets total, $n = 3$ donors). **b** Individual traces from four representative delta cells within a single islet in vivo and showing the profile of spiking and slow delta cells. Insulin (0.25 U/kg) was injected 2 min prior to glucose injection (0.4 g/kg i.v.). Gray-shaded area indicates a predetermined time zone used for data analysis and marks the glucose stimulation period. Basal and recovery periods are located to the left and right of the grey-shaded area, respectively. **c** Relative distribution (or frequency) of detected peaks in all spiking delta cells analyzed ($n = 217$) over the course of up to 18 min. Grey-shaded area represents the glucose-stimulated time period as described in **b**. **d** Normalized spiking delta cell activity peak amplitude for basal ($n = 4574$ peaks); glucose ($n = 7850$ peaks); and recovery ($n = 10518$ peaks) time periods. In **e**, same as in **d**, however the graph shows the breakdown of the normalized peak amplitude in the first or last 2 min of the glucose stimulation time period. **f** Hierarchical clustering analysis of the spiking delta cell activity under basal, the first or last 2 min of glucose-stimulated or recovery time periods. **g** Normalized delta cell peak amplitude in nonsynchronized (OFF) and synchronized (ON) peaks. In **a**, scale bar = 35 μm. Statistics: in **c**, \*$p = 0.0001$ vs time 0 (**d**, **e**, and **g**) \*\*$p < 0.0001$, (**g**) #$p < 0.0001$ vs respective OFF group. All comparisons made with OneWay Anova with a Tukey multiple comparison test. Error bars represent the 95% confidence interval (C.I.) of the mean. For **a**, **c–e**, and **g**, source data are provided as a Source Data file

islets delta cell filopodia allowed an ~ tenfold increase in potential direct interactions with beta and alpha cells (alpha, from 0.69 ± 1.49% to 6.3 ± 10.67%; beta, 0.48 ± 0.99% to 6.26 ± 10%, Fig. 2e). These data confirm that the filopodia are a potential avenue of communication between delta cells and other neighboring endocrine cells.

**Delta cell filopodia are dynamic**. The observed variations in filopodial length (Fig. 2d) suggested that delta cell morphology could be dynamic. We therefore used TPM to image, over spans of hours (Fig. 2f) and days (Fig. 2g), single delta cells in vivo in SST-ChR2-YFP islets transplanted into the ACE. To determine the relative position of individual delta cells, we labeled the islet vasculature with intravenously injected dye (150 kDa TRITC-dextran) and then used the labeled islet capillaries as fiducial markers[30]. We observed that delta cells were either positioned directly along capillaries or were connected to the vascular network via filopodia (Fig. 2f, g). Filopodia-dependent connections between delta cells and islet vasculature occurred in situ (Supplementary Fig. 4A, Supplementary Movie 4) and were stable for days in vivo (Fig. 2g). However, filopodia not connected to the vasculature were highly dynamic, exhibiting noticeable filopodia extension, and repositioning over timescales as fast as 30 min (Fig. 2f, Supplementary Fig. 4). These observations show that delta cells can rapidly reposition their filopodia within the intra-islet milieu, suggesting that the somatostatin signaling landscape within the islet is dynamic.

We next considered the chemical signals that could drive filopodia dynamics and thereby guide them towards their target cells within the islet by considering three factors, namely insulin, IGF-1[31] or VEGF-A[32]. Insulin was chosen because it is secreted in abundance by beta cells, while VEGF-A was chosen due to the fact that it is a beta cell-secreted factor that acts on endothelial cells, which are in turn targeted by the delta cell filopodia (Fig. 2g, Supplementary Fig. 4). Importantly, delta cells express insulin and IGF-1 receptors (IGF-1R)[11]. Although incubating isolated mouse islets for 24 h with high insulin (1 μM) may lead to a decrease in filopodia length ($p = 0.0284$, FDR > 0.05), treatment with IGF-1 or VEGF-A did not noticeably affect filopodia length (Fig. 2h). However, blockade of endogenous islet IGF-1 receptors[11] with picropodophyllin (PPP) or of VEGF-A receptors[11] with axitinib (Axtb) significantly decreased filopodia length (Fig. 2h). While these effects could not be reversed by IGF1 or VEGF-A treatment, incubation with 1 μM insulin did restore normal filopodia length in PPP-treated islets (Fig. 2h, see Supplementary Data 1–2 for detailed statistics). These data suggest that endogenous VEGF, IGF1, and insulin receptor signaling, in states of acute hyperinsulinemia, can stimulate elongation of delta cell filopodia.

**Secretory capacity and $[Ca^{2+}]_i$ dynamics of delta cell filopodia**. We next asked whether delta cell filopodia have a secretory/regulatory role. From our in vivo TPM imaging dataset (Fig. 1), we identified a total of five delta cells with discernable filopodia in four different SST-GCaMP3 islets ($n = 1$ islet/mouse, total of four different mice). Under resting conditions, delta cell somata and filopodia displayed low amplitude $[Ca^{2+}]_i$ spikes that were not synchronized, with the filopodia displaying generally lower amplitudes (Fig. 3a). Analyses of the first 80 s after glucose injection revealed a marked increase in $[Ca^{2+}]_i$ spiking frequency and spike amplitude in both cell compartments (Fig. 3a–f). In some cases, filopodia exhibited $[Ca^{2+}]_i$ spikes that were not synchronized with somatic $[Ca^{2+}]_i$ spikes (Fig. 3b, c). Because glucose stimulation of somatostatin release depends on an increase in $[Ca^{2+}]_i$[17], these results indicate that somatostatin secretion may occur independently from both the soma and filopodia of delta cells. Another requirement for somatostatin release from filopodia is $[Ca^{2+}]_i$-dependent membrane trafficking and vesicular exocytosis followed by membrane recycling via endocytosis. We incubated isolated SST-ChR2-YFP islets with the reporter dye FM-4-64 (Fig. 3g–i), a highly fluorescent membrane probe that labels internalized secretory vesicles[31]. We observed FM-4-64 accumulation in the delta cell filopodia within minutes (Fig. 3g–i), indicating active recycling of exocytotic vesicles. Furthermore, several components required for SST secretion, including the L-type $Ca^{2+}$ channel subunit CaV1.2 (Supplementary Fig. 5A–D), vesicle-associated membrane protein-2 (VAMP2) and synaptophysin (Supplementary Fig. 5E–T) were observed in the filopodia in mouse and human delta cells.

We investigated the ultrastructure of delta cell filopodia at nanometer resolution by performing multi-tilt electron tomography on a 250-nm thick section of a mouse islet. We identified two examples of delta cell filopodia (Fig. 3j and Supplementary Fig. 6A) and performed 3D reconstruction of one delta cell filopodium. The first filopodium contained granules localized in close proximity to both the plasma membrane and a microtubule (Fig. 3j, k). The second filopodium had fewer granules and contained mitochondria in the upper half, while the lower half lacked granules and contained endoplasmic reticulum (Supplementary Fig. 6A). We confirmed the occasional existence of delta cell filopodia lacking somatostatin in both mouse and human islets with immunohistochemistry (Supplementary Fig. 6B–J). These structures likely represent intermediates in the elongation (or retraction) of delta cell filopodia. Together, our in vitro and in vivo data support that the delta cell filopodia have dynamic $[Ca^{2+}]_i$ spiking behavior and somatostatin granules fitted with elements of an active exocytosis machinery, consistent with a possible role for filopodia in somatostatin release (Fig. 3, Supplementary Figs. 5 and 6).

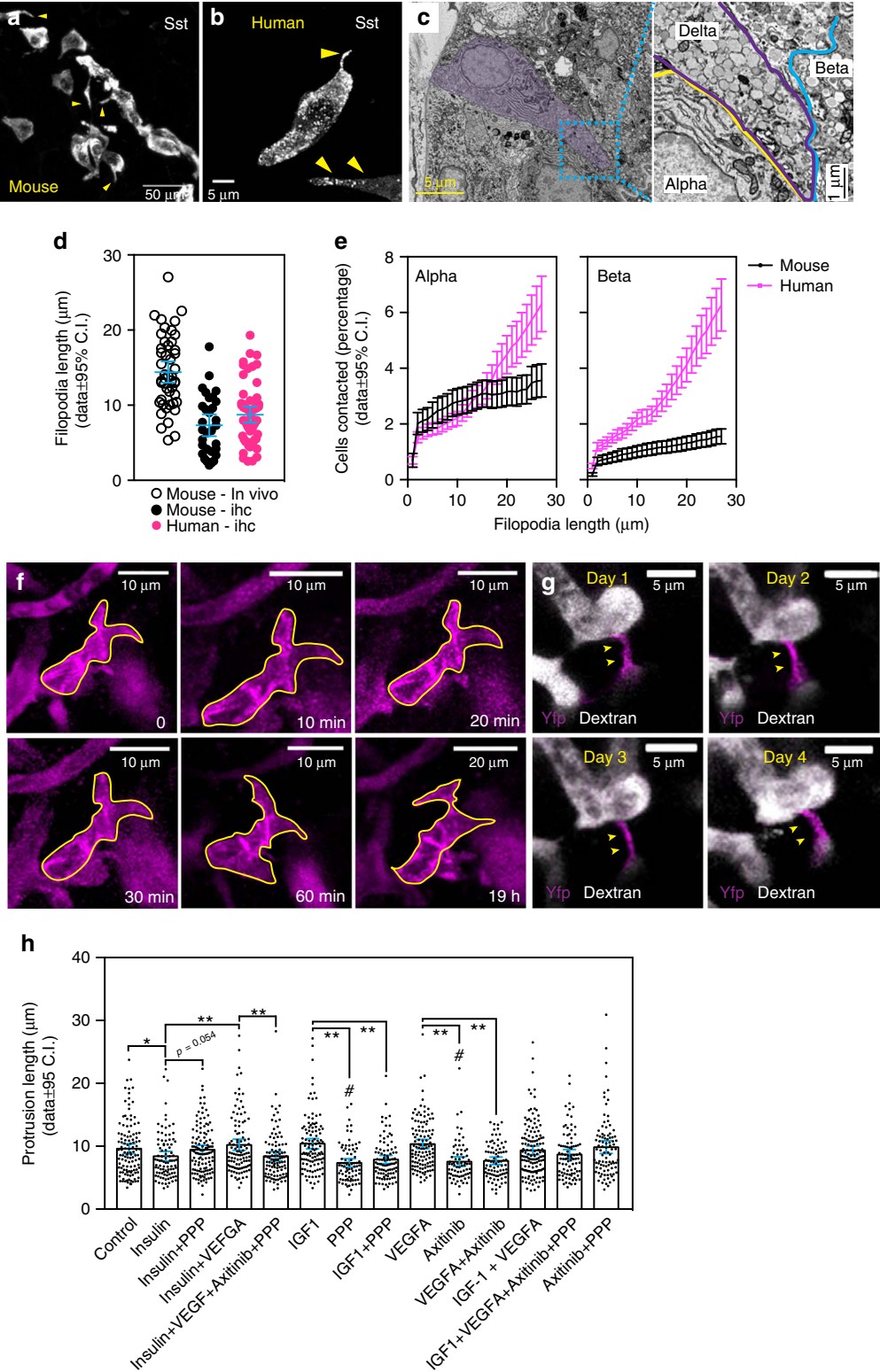

## In vivo optogenetic activation of delta cells and beta cell [Ca²⁺]ᵢ imaging

**In vivo optogenetic activation of delta cells and beta cell [Ca$^{2+}$]$_i$ imaging.** To determine whether somatostatin release from delta cells exerts paracrine control over beta cells, we used ChR2 to photostimulate delta cells in vivo. In SST-ChR2-YFP islets, ChR2-YFP was expressed in ~80% of delta cells[33] (Supplementary Fig. 7A–D) and virtually all ChR2-YFP$^+$ cells expressed somatostatin (Supplementary Fig. 7E). SST-ChR2 mice had normal glucose homeostasis (Supplementary Fig. 7F). In

pancreas slices, illumination of ChR2-expressing delta cells with a 470 nm light beam evoked an inward photocurrent (Fig. 4a; peak amplitude 273 + 48 pA at −70 mV; $n = 12$). The exact impact of photostimulation on membrane potential changes in the intact cell, being spontaneously active, is not known. Nevertheless, optogenetically evoked depolarizing currents were sufficient to obtain voltage changes mimicking the action potentials of the delta cell (Fig. 4a) and similar to those evoked by depolarizing

**Fig. 2** Cyto-architecture of delta cells in rodents and human islets. Delta cells send out cellular processes (filopodia) to increase their intra-islet reach. Maximum projection images of representative mouse (**a**) and a human delta cells (**b**) immunostained with anti-somatostatin antibodies. The yellow arrowheads indicate cellular extensions from delta cells. **c** Scanning electron microscopy of a human islet. Somatostatin-expressing delta cell is highlighted by a purple shade. Right, close-up image of the delta cell filopodia (highlighted by the cyan box in **c**) contacting an alpha and a beta cell. **d** Physical length of the delta cell filopodia in its longest axis (tip to cell body) in vivo in mouse islets (empty black circles, $n = 47$ cells) and in PFA-fixed pancreases from mice (black circles, $n = 31$ cells) and human (pink circles, $n = 57$ cells) samples. **e** Relative reach of single delta cells towards alpha and beta cells in mouse (black line) and human islets (purple line). **f** In vivo longitudinal follow up of a single delta cell at 10-min intervals or 19 h later. Delta cells were from SST-ChR2 islets and imaged with TPM to avoid rhodopsin activation[28]. A yellow line highlights the delta cell morphology over time. **g** In vivo longitudinal follow up of a single delta cell filopodium over the course of 4 days as in **f**. Vessels were visualized by an i.v. injection of TRITC-labeled dextran molecules. Yellow arrowheads indicate the filopodium from the same delta cell from days 1 to 4. **h** Delta cell filopodia length from isolated islets treated with insulin, IGF, and VEGF-A with or without the IGF-1R receptor blocker picropodophyllin (PPP) or the VEGFR blocker Axitinib. Statistics shown for FDR ($q < 0.005$ for discovery): *$q = 0.052$, $p = 0.0284$; **$q < 0.002$, $p < 0.001$, #$q < 0.002$, $p < 0.001$ vs control (DMSO) group. Detailed statistics shown in Supplementary Data 1. Error bars represent the 95% confidence interval (C.I.) of the mean. For **d**, **e**, and **h**, source data are provided as a Source Data file

current pulses (Supplementary Fig. 7G). Hence, light could evoke trains of action potential-like responses at frequencies up to 5 Hz, as well as maximal somatostatin release[34]. Next, we combined delta cell optogenetics with in vivo beta cell $[Ca^{2+}]_i$ imaging by transducing isolated control or Sst-ChR2 islets with vectors encoding the $Ca^{2+}$ reporter GCaMP6[35] under the control of the insulin promoter. Sst-ChR2/Ins-GCaMP6 islets were transplanted into the ACE of host mice and imaged under nonfasting conditions with or without light stimulation (methods, Fig. 4b, c). Without light stimulation, beta cells in both control and SST-ChR2 islets displayed $[Ca^{2+}]_i$ oscillations (Fig. 4d). Such oscillations were more frequent in ChR2 animals (Fig. 4b), likely due to changes in basal delta cell activity by exposure to ambient light prior to photostimulation. Upon light stimulation (four pulses, each 15 s, details in the "Methods" section), beta cells of WT islets responded heterogeneously, with either no change or a slight increase in $[Ca^{2+}]_i$ oscillation amplitude (Fig. 4e). This effect on $[Ca^{2+}]_i$ oscillations is likely due to light activating the pupillary reflex and causing parasympathetic release of acetylcholine, which enhances beta cell function[4,36]. In light-stimulated SST-ChR2 islets, most beta cells displayed reductions in $[Ca^{2+}]_i$ oscillation, amplitude, and frequency (Fig. 4e, f). On average, there was a 35% decrease when compared with nonstimulated conditions (Fig. 4e, f), consistent with the reported transient inhibitory effect of somatostatin on beta cell electrical activity[37,38].

**Delta cell activity is impaired in an animal model of prediabetes.** Earlier studies have suggested that delta cell activity and/or somatostatin release is impaired in islets exposed in vitro to free fatty acids[39] and in animal models of prediabetes and diabetes[40,41]. Prediabetes is a state where higher insulin demands are compensated for by increased beta cell secretory activity, followed by an adaptive proliferation of beta cells. To determine the potential impact of prediabetes on delta cell function, mice with SST-GCaMP3 islets engrafted for 1 month were fed a chow (CD) or a high-fat diet (HFD) for 8 weeks. HFD-feeding impaired glucose homeostasis and elevated fasting glucose levels (Supplementary Fig. 8A, B). This phenotype was associated with a trend towards increased beta cell number without any changes in total islet area (Supplementary Fig. 9A, B), indicating that our HFD feeding model recapitulates a prediabetes state that was maintained for the duration of our in vivo imaging experiments (Supplementary Fig. 8C). In control animals, glucose stimulation significantly increased delta cell $[Ca^{2+}]_i$ spike amplitude (Fig. 5a, b) but not spike frequency or cell synchronization 3 months after ACE transplantation (Supplementary Fig. 8F–G). HFD-fed mice exhibited a marked impairment of delta cell activity in response to glucose stimulation, characterized by overall lower $[Ca^{2+}]_i$ peak amplitudes, a higher number of unresponsive cells, and a

doubling of the number of oscillatory cells that were also functionally impaired (Supplementary Fig. 8D, E).

Next, we tested whether delta cells in HFD-fed animals displayed significant morphological changes when compared with CD-fed islets and found that HFD feeding was associated with an increase in delta cell filopodia length (Fig. 5c). As a result, the total number of beta cells in direct contact with individual delta cells, or with the entire delta cell population, was similar between CD- and HFD-islets (Supplementary Fig. 9C, D). In addition, we observed a similar contact density between delta and beta cells when we simulated the contribution of the delta cell filopodia (Supplementary Fig. 9E, F).

These results indicate that delta cell function is impaired during the onset of hyperglycemia in prediabetic animals, a physiological condition associated with beta cell expansion and $[Ca^{2+}]_i$ dysfunction[21]. Of note, the longer filopodia observed in prediabetic delta cells may reflect adaptation to maintain contact with beta cells in prediabetic mice (Fig. 5c, Supplementary Fig. 8C–F).

**Discussion**

Delta cells are important paracrine regulators of alpha and beta cell function[13] and inhibit their activity (Fig. 4c–e) by secreting somatostatin in response to paracrine or circulating stimuli[11,15–17,42]. Consequently, delta cells can indirectly affect the control of glucose homeostasis in health and disease[43]. However, it has been largely unknown how delta cells are able to effectively regulate the function of alpha and beta cells while being a relatively rare cell type among the islet endocrine cell population (Fig. 1a [1,3]). Furthermore, it has been suggested that diabetes is associated with impaired somatostatin release and/or delta cell death[40,41,44].

We now show that the somatostatin-secreting delta cell is able to rapidly inactivate beta cells in vivo, which serves as an efficient brake to prevent undesirable and deleterious overshooting of insulin release. In vivo, delta cells are active under resting and glucose-stimulated conditions and are rapidly inactivated after a burst of glucose-induced activity. Besides glucose[17,18], activation of delta cells can also be mediated by insulin secreted from neighboring beta cells in response to rising glucose levels[45]. Delta cell filopodia are adapted to contact neighboring alpha and beta cells, a phenomenon that is maximized in the human islet and dependent on endogenous IGF1, insulin and VEGF signaling pathways. Islet endocrine cells have been reported not to express VEGFRs[11]. Hence, it is likely that the filopodia phenotype observed after inhibition of VEGF signaling pathways is mediated indirectly by islet endothelial cells, which remain in isolated islets for days[46]. Filopodia are dynamic and somatostatin release competent, allowing the delta cell to rapidly target different regions within the islet. Furthermore, prediabetic conditions are sufficient to induce a state of hyperglycemia that is associated with

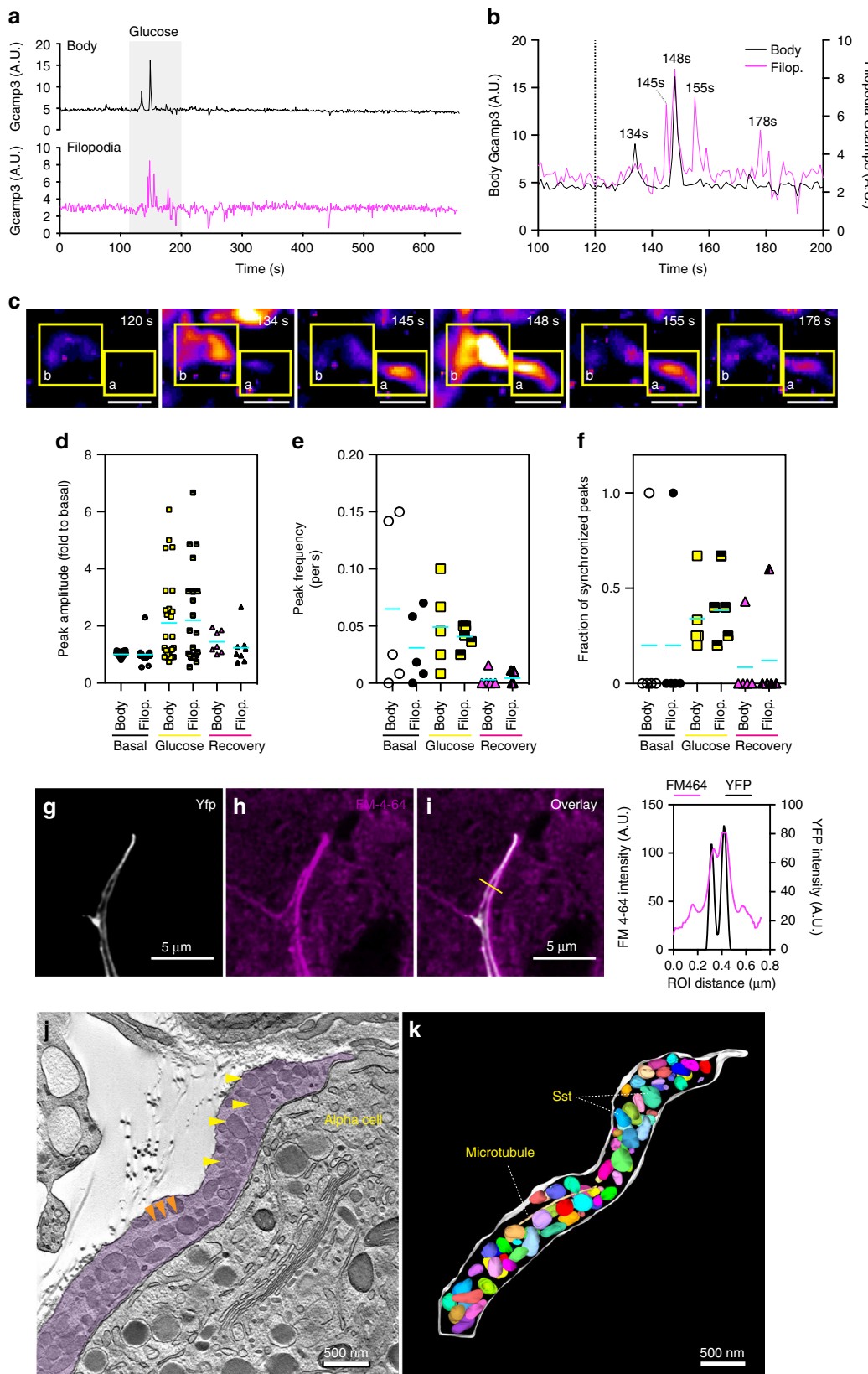

impairment of delta cell function and altered morphology. The timescale of delta cell dysfunction parallels that of beta cells under similar prediabetic conditions, where higher circulating insulin and somatostatin levels should occur due to both increased basal beta cell secretory activity[21] and a direct stimulatory effect of insulin on the delta cell[45]. Our results thus provide insight into the activity of the delta cell in health and prediabetes and a possible mechanism for somatostatin to effectively exert its potent suppressive effects on islet beta and alpha cells and thereby act as an efficient modulator of glucose homeostasis.

**Fig. 3** Islet delta cell filopodia have independent Ca$^{2+}$ spikes from the soma and contain secretory granules. Delta cell filopodia display Ca$^{2+}$ spikes in vivo upon stimulation with glucose. **a** Representative Ca$^{2+}$ dynamics of a delta cell soma (black line) and filopodia (pink) in vivo. Shaded grey area indicates the time point when glucose (0.4 g/kg) was injected i.v. **b** Close-up of the area highlighted in grey in **a**, glucose stimulation time point is indicated by a dashed line. Numbers on top of the peaks indicate the time point of the peak's maximum amplitude. Values for the cell body are plotted on the left *Y*-axis while values for the filopodia are plotted on the right *Y*-axis. Dotted line indicates the time point when glucose was injected. **c** Maximum projection images of Ca$^{2+}$ dynamics in delta cell body and filopodia, shown in **a**, **b**. Yellow squares mark the regions of interest represented in **a**, **b**. **d** Peak amplitude and **e** frequency in delta cells (*n* = 5) with soma and filopodia regions at basal, glucose, and recovery time points. **f** Peak synchronization between the delta cell soma and filopodia at basal, glucose, and recovery time points. **g–i** Single confocal slice of a delta cell filopodia from an SST-ChR2 islet incubated with FM-4-64 at 3 mM glucose. The delta cell filopodia plasma membrane was visualized by imaging YFP (**g**), FM-4-64 is shown in **h** and the overlay is shown in **i**. Graph on the right of **i** indicates the pixel profile of the region of interest (yellow line) drawn in **i**. YFP pixels (plasma membrane) are represented by a black line and plotted on the right *Y*-axis. FM-4-64-pixel profile is shown in pink and plotted on the left *Y*-axis. **j** Electronic tomogram of a delta cell filopodia in a mouse islet, cytoplasm is highlighted by a purple shade. Orange arrowheads indicate a microtubule and yellow arrowheads point to somatostatin granules. **k** 3D reconstruction of the volumetric data set shown in **j**. Filopodia plasma membrane is shown in white and somatostatin granules are shown in varying colors. Scale bars, **c** 10 μm, **g–i** 5 μm, and in **j**, **k** 500 nm. Error bars represent the 95% confidence interval (C.I.) of the mean. For **d–f** source data are provided as a Source Data file

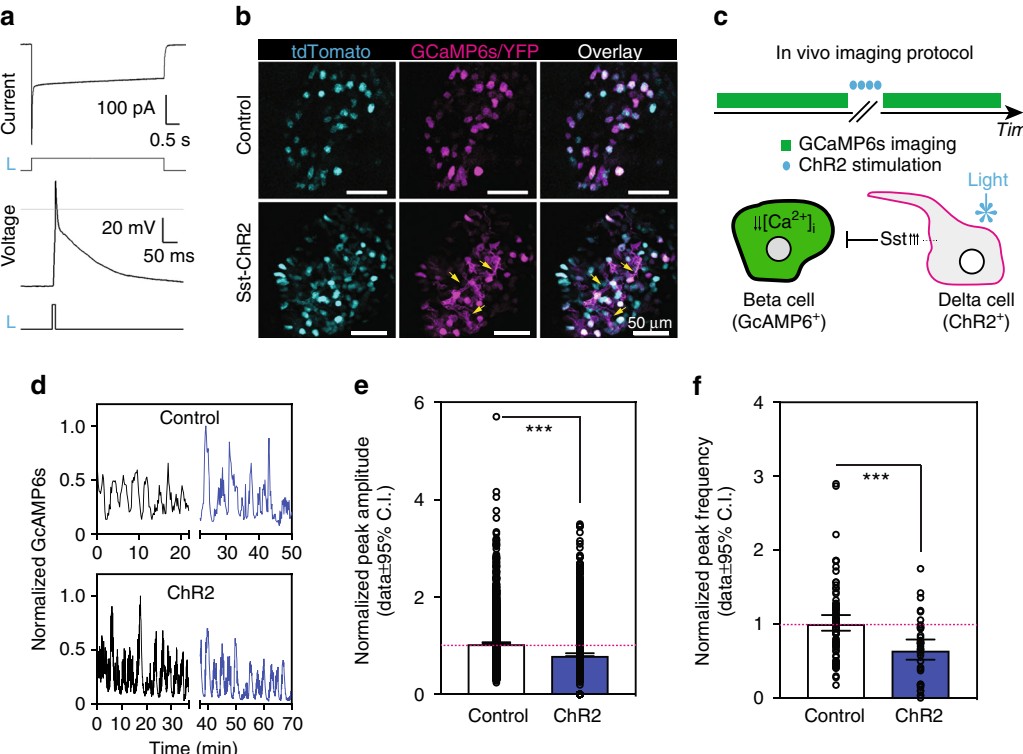

**Fig. 4** Optogenetic control of delta cell activity. Somatostatin reduces beta cell [Ca$^{2+}$]$_i$ oscillatory frequency and peak amplitude, and hyperglycemia is associated with impaired delta cell function and longer filopodia. **a** Photostimulation of delta cells expressing ChR2 evoked inward photocurrents under voltage clamp conditions (top; holding potential = −70 mV) and light-evoked, overshooting action potential-like responses under current-clamp conditions (bottom). Timing of light flashes (470 nm) is indicated by lower traces (L). Dashed line = 0 mV. **b** tdTomato and GCaMP6s/YFP fluorescence signal in wild-type and SST-ChR2 islets. **c** Top, protocol for in vivo imaging and ChR2-mediated photostimulation (450–470 nm light). Bottom, cartoon illustration representing the experimental setup and expected biological responses of GCaMP6$^+$ beta cells and ChR2$^+$ delta cells. **d** Representative beta cell [Ca$^{2+}$]$_i$ traces in islets from wild-type (top) or SST-ChR2 mice (bottom). **e** Normalized beta cell [Ca$^{2+}$]$_i$ peak amplitude and **f** peak frequency in wild-type (*n* = 7 islets) and SST-ChR2 islets (*n* = 5 islets) before and after ChR2 stimulation in vivo. **e** ***$p < 0.001$ by two-tailed student *t*-test and in **f**, ***$p = 0.001$ vs light-stimulated WT islets. In **e**, **f** data are displayed as fold change from nonstimulated conditions, represented by the horizontal pink dotted lines. Error bars represent the 95% confidence interval (C.I.) of the mean. For **e**, **f** source data are provided as a Source Data file

## Methods

**Animals**. All animal procedures were approved by the Institutional Animal Care and Use Committee (IACUC, protocol number 2013/SHS/816) of the SingHealth system or the Karolinska Instituet (protocol number N34/16) and complied with all relevant ethical regulations for animal testing and research. Transgenic mouse strains used in this study were from Jackson Laboratories. For in vitro studies, male and female mice were used. For in vivo imaging, female mice were used as donors and recipients. The following strains were used: SST-CRE (Ssttm2.1(CRE)$^{Zjh/J}$), Ai32-ChR2-YFP (B6;129S-Gt(ROSA)26$^{Sortm32(CAG-COP4*H134R/EYFP)Hze/J}$), and

GCaMP3 (B6;129S-Gt(ROSA)26Sor$^{tm38(CAG-GCaMP3)Hze/J}$). Wild-type (WT) C57/6NTac (In Vivos, Singapore) or C57/BL6J (Charles River, USA) mice were used as hosts of Sst-ChR2 and Sst-GCaMP3 islets, respectively. Glucose tolerance tests (GTT) were performed as described in[30]. Briefly, animals were fasted for 4–6 h after which the fasting blood glucose levels were measured at two time points 5 min apart. Next, mice were injected with 2 g glucose/kg body weight and blood glucose levels were measured every 15–30 min. For I.V.GTT in Sst-GCaMP3 transplanted mice, animals were anesthetized according to the imaging protocol (i.e., i.p. injection of a mixture of fluanisone (25 mg/kg), fentanyl (0.788 mg/kg) and

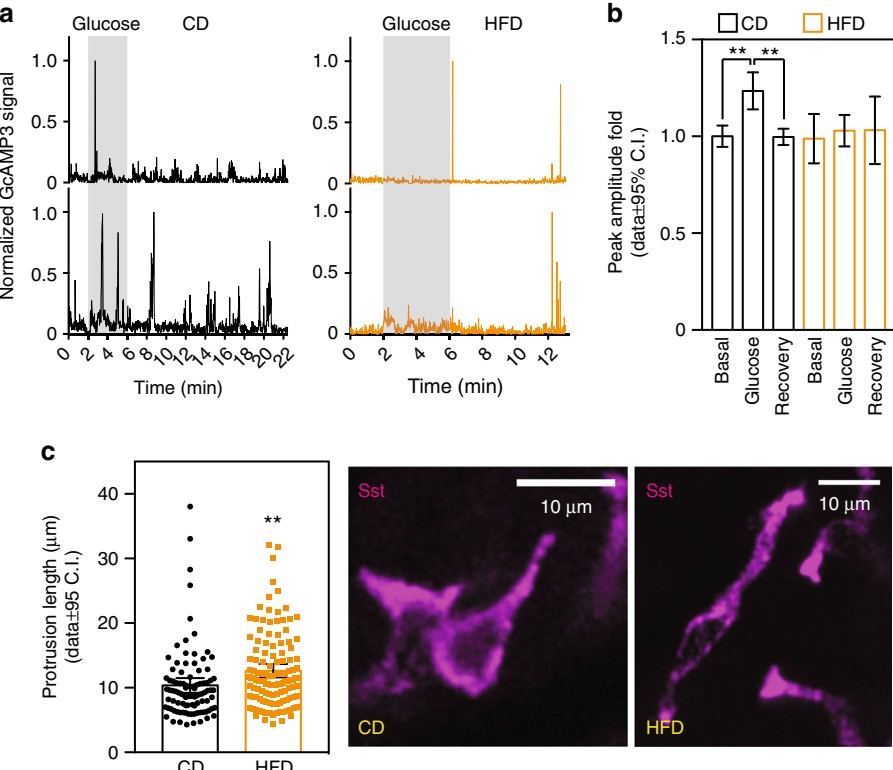

**Fig. 5** Delta cell activity is impaired in prediabetes. **a** Representative normalized traces of GCaMP3 signals measured in vivo in delta cells from animals fed with control (CD, black lines) or HFD (orange lines). As in Fig. 1, the yellow-shaded area marks the time period used for measurements of glucose responses. **b** Spiking delta cell $Ca^{2+}$ peak amplitude signal during basal, glucose, or recovery time periods normalized to basal conditions in CD (basal: $n = 358$; glucose: $n = 588$; recovery: $n = 1060$ peaks from $n = 23$ cells in four islets total and one islet per animal) or HFD (basal: $n = 127$; glucose: $n = 224$; recovery: $n = 353$ peaks from $n = 14$ cells in four islets total and one islet per animal for a total of four animals) animals. Data shown as fold change from the basal state. **c** Physical length of the delta cell filopodia in its longest axis (tip to cell body) from islets in situ from CD ($n = 4$ mice, 33 islets total, $n = 33$ delta cells) or HFD ($n = 4$ mice, 42 islets total, $n = 42$ delta cells) animals. Right, maximum projection images of delta cells acquired with confocal microscopy in CD or HFD animals. Scale bars, **a** 50 μm and (H), 10 μm. Statistics: **b** data shown for false discovery rate (FDR, $q < 0.005$ for discovery): **$q < 0.002$, $p < 0.001$ by OneWay Anova with a multi-comparison test using a two-stage linear step-up procedure of Benjamini, Krieger, and Yakutieli. In **c**, **$p = 0.0038$ by unpaired, two-tailed Student's $t$-test. Error bars represent the 95% confidence interval (C.I.) of the mean. For **b**, **c** source data are provided as a Source Data file

midazolam (12.5 mg/kg)), glucose was injected in the tail vein (0.4 g/kg) and blood glucose was measured every 2 min (Accu-Check, Roche, Switzerland). For dietary treatments, animals were assigned for 6–7 weeks to either a control diet, CD (CD88137, 11% of energy from fat, 66% from carbohydrates, and 23% from proteins) or to a HFD (TD88137, 43% of energy from fat, 42% from carbohydrates, and 15% from proteins). Diets were acquired from Ssniff Spezialdiäten GmbH, Soest, Germany.

**Transplantation of islets into the anterior chamber of the eye (ACE) and in vivo microscopy.** Transplantation and in vivo imaging of ACE islets were performed as previously described[20]. In brief, islets from SST-ChR2-YFP or SST-GCaMP3 mice were isolated 24–48 h prior to transplantation using the collagenase method and recovered in vitro with fresh CMRL-1066 or RPMI 1640 medium supplemented with 10% fetal bovine serum. Delta cell GCaMP3 imaging was performed under the following anesthesia conditions: mixture of fluanisone (25 mg/kg), fentanyl (0.788 mg/kg), and midazolam (12.5 mg/kg). To image delta cell GCaMP3 in vivo (data shown in Fig. 1), animals were injected with insulin (0.25U/kg) 2 min prior to the injection of glucose. We determined that this 2-min time window would be used as a baseline for calculating the effect of glucose stimulation on delta cell activity. Subsequent imaging of delta cell GCaMP3 in animals fed CD or HFD diets did not involve insulin injection prior to glucose stimulation. When beta cell function was investigated in WT or SST-ChR2-YFP islets with GCaMP6s constructs, islets were transduced with an adenovirus carrying a RIP2-GCaMP6s-IRES-tdTomato construct immediately after isolation and the media was changed 24 h prior to transplantation. RIP2-GCaMP6s-IRES-tdTomato adenoviruses were produced in HEK293 cells and harvested 4–6 days after initial transfection. Next, 20 μL of the media containing the viruses was added to a 6-mm dish containing the freshly-isolated islets in 2 mL of RPMI 1640 medium

supplemented with 10% fetal bovine for 16 h. The next day, the media was changed and islets incubated for 6 h prior to transplantation. The efficiency of this transduction protocol was variable ($n = 2$–50 beta cells GCaMP6s-positive 2 days after transplantation) and the beta cells transduced were limited to the outer layers of the islet. On average, 5–20 islets were transplanted into the right ACE of the host mouse (8–12 weeks old) under anesthesia (2% isoflurane) as in reference[20]. In vivo imaging was performed with an upright multiphoton SP5 or SP8 microscope. For imaging delta cell function (SST-GCaMP3 mice), islets were allowed to engraft for at least 30 days prior to imaging with single- and two-photon laser scanning microscopy. For imaging beta cell function (WT or SST-ChR2-YFP islets transduced with the RIP2-GCaMP6s-IRES-tdTomato construct), islets were engrafted for 10–12 days prior to start of the experiments. In vivo optogenetic stimulation of delta cells was performed by pulsing ($4 \times 15$-s pulses) a blue light (450–490 nm) through the imaging objective focused on the engrafted islet (Fig. 4a). For monitoring of delta cell filopodia movement, SST-ChR2-YFP islets were imaged 90 days after transplantation. To avoid uncontrolled activation of delta cell ChR2 during imaging, islets and delta cells were imaged with two-photon LSM using a bidirectional resonant scanner (8000 Hz, Ex 970 nm Em 500–550 nm), that does not activate ChR2[28]. Imaging of GCaMP6s and tdTomato in controls and SST-ChR2-YFP islets was achieved with a laser wavelength of 1000 nm. Beta cells were identified by green (GCaMP6s) and red (tdTomato) fluorescence signal. ChR2-YFP delta cells were identified by a weak green (YFP) fluorescence. Eventual drifts in X, Y, and Z were corrected post hoc with custom registration algorithms described in ref. [47].

**Islet immunohistochemistry.** Immunohistochemistry of pancreas and isolated islets was performed in 30–40 μm thick pancreas slices or isolated islet whole mounts as described before[30]. In brief, mouse and human pancreata were fixed overnight in fresh 4% paraformaldehyde and cryo-protected in 30% sucrose for

12 h and embedded in OCT and sectioned. Human pancreas was from cadaveric donors ($n = 5$, Singapore General Hospital). Use of human tissues (with no traceable information besides age and sex) was approved by the SingHealth Centralized Institutional Review Board (IRB#2013-504-A), complied with all relevant ethical regulations for use of human tissues in research and had donor consent for use in research and publication. The following primary antibodies were used: guinea pig anti-insulin (pancreas 1:400, isolated islets 1:1500, Dako - A0564), mouse anti-glucagon (1:400, Sigma-Aldrich—G2654), rat anti-somatostatin (pancreas, 1:400 Millipore—AB5494, isolated islets 1:700, BioRad 8330-0009), goat anti-CD31 (1:100, RD Systems—AF3628), rabbit anti-VAMP2 (1:100, Cell Signaling—13508 S), rabbit anti-synaptophysin (1:100, Abcam—ab32594), and rabbit anti-CaV1.2 (1:50, Alomone—ACC-003). Secondary antibodies were raised in donkey and were conjugated to Alexa 488, 546, 561, or 647 fluorophores. Nuclei were labeled with DAPI (1:400, Invitrogen). Incubation of isolated SST-ChR2 islets with FM-4-64 (Invitrogen, T13320) was performed at room temperature (~23 ℃), according to manufacturer guidelines. Islets were incubated with FM-4-64 at low glucose (3 mM) in $Ca^{2+}$ buffered solution (125 mM NaCl, 5.9 mM KCl, 2.56 mM $CaCl_2$, 1.2 mM $MgCl_2$, and 25 mM HEPES) for 3 min the day after isolation and overnight recovery in vitro. After incubation, islets were fixed in 4% PFA for 30 min, mounted on a coverslip and prepared for imaging.

**Simulation of delta cell contact densities and reach in islet sections**. Mouse and human islets immunostained with anti-insulin, -glucagon, and/or -somatostatin antibodies and DAPI were mapped using an imageJ plugin as previously described[29]. In brief, the nuclei and cells are segmented in 3D, which generates a "geographical" map of alpha, beta, and/or delta cells in XY and Z planes of each islet (Supplementary Fig. 3A–C). Next, we determined the number of direct delta cell contacts (characterized by adjacent cell boundaries[29]) with neighboring alpha or beta cells. This information is translated into a measure of "contact frequency", which is the percentage of alpha or beta cells contacted by a given delta cell (e.g., if a delta cell contacts 2 out of 20 total mapped alpha cells, that delta cell's contact frequency with alpha cells is 10%). This process is repeated for each delta cell mapped. Next, once the position of each alpha and/or beta and delta cell is known, we simulate the expected contact density that would be observed if the boundaries of each delta cell were expanded by 1 μm increments in all possible directions (except towards the outside of the islet structure) until it reaches a maximum "filopodia length" value determined by the user (Supplementary Fig. 3J).

**In vitro islet treatment with axitinib, picropodophyllin, or insulin**. Isolated islets were used 6 h after isolation and incubated with the VEGFR inhibitor (axitinib, Sigma-Aldrich, 0.1 μM) and/or the IGF-1R inhibitor (Picropodophyllin, PPP, Sigma-Aldrich, 0.1 μM) for 30–60 min prior to incubation with VEGF-A (Pepro-Tech #450-32, 100 ng/mL), IGF-1 (Novus Biologicals 791-MG, 20 ng/mL) or insulin (NovoNordisk, 1 μM) for the following 24 h. After incubation, isolated islets were fixed in 4% PFA for 30 min and immunohistochemistry was performed as described above. Measurements of delta cell filopodia for all experimental groups were performed in a blind fashion, where the person analyzing the images did not know the identity of all experimental groups until the analysis was completed.

**High-resolution confocal imaging of pancreatic islets and imaging processing**. Confocal imaging was performed on a Leica up-right or inverted SP5 or SP8 microscope fitted with a white light laser and gated Hybrid detectors. FM-4-64 stained islets were imaged using the following excitation and emission configuration: YFP, Ex510nm and Em520-570nm, FM-4-64, Ex510nm and Em630–680nm. Islet viability was checked prior to imaging with backscatter analyses[20]. For co-localization studies, isolated islets or sections were mounted on 80% glycerol/PBS (with 0.2% n-propyl gallate (NPG, Sigma) as anti-fade agent) and acquired with a glycerol 63×/1.3NA objective with a correction collar adjusted to the focal plane prior to imaging. Images were acquired using Nyquist acquisition parameters for the X,Y, and Z axis and de-convolved using Huygens software (Scientific Volume Imaging). For deconvolution, the signal-to-noise ratio was determined for each channel and the deconvolution parameters set for 40 iterations and a quality threshold of 0.05. Co-localization was determined with IMARIS software (Bitplane) co-localization module and automatic thresholding for each channel. Imaging of mouse pancreatic sections from animals fed with a CD or HFD was performed in samples mounted in Prolong® antifade solution with DAPI (ThermoFisher) and acquired with a glycerol 100× objective. Imaging of isolated islets treated in vitro, was performed in PBS with a water-immersion 25×/0.95NA objective. Somatostatin was imaged with Alexa488 secondary antibodies and TPM with an excitation line of 800 nm. For simulations determining the reach and contact density of delta cells in mouse and human islets, we utilized a 3D tissue analysis ImageJ algorithm as previously described[29].

**In vivo $Ca^{2+}$ trace imaging data extraction and analysis**. Extraction of in vivo delta cell $Ca^{2+}$ traces (i.e., raw GcAMP3 signal) was done in MATLAB using the CaImAn plugin as previously described[48]. In brief, we used the "demo script" within the CaImAn package to extract the $Ca^{2+}$ data from movement-corrected and background-subtracted GCaMP3 image stacks. We used the script to identify cells (i.e., regions of interest, ROIs) automatically (CaImAn parameters: $K = 50$, tau = 10, and $p = 1$) and normalized the GCaMP3 signal to the highest intensity peak in each cell. As a second step, the software-detected ROIs were manually curated to make sure that only ROIs representing cells were selected for analysis. Next, we ran a peak-finding function and considered all peaks that met the arbitrary threshold of 1× higher than the standard deviation of the whole in vivo recording for each cell. This threshold was chosen because it could detect peaks in all control and HFD-fed animals imaged in vivo, where the latter have overall lower peak amplitudes that could not be detected with higher thresholds. Next, the peak location and amplitude for each cell was plotted and analyzed using excel software. In addition, the modified demo script contains a classic MATLAB function that integrates the normalized GCaMP3 signal from each detected cell in a given islet to generate a hierarchical clustering analysis (HCA) (details in https://www.mathworks.com/help/stats/hierarchical-clustering.html). Analysis of beta cell $Ca^{2+}$ traces acquired via GCaMP6s imaging was done as follows: First, we calculated the GCaMP6s/dtTomato signal intensity ratio for each identified beta cell for the entire imaging session. Next, the files containing the GCaMP6s/dtTomato ratio were analyzed with a custom MATLAB script that identified $[Ca^{2+}]_i$ peaks that passed the arbitrary threshold of 1.5× above the standard deviation for the entire imaging experiment and for each cell. Next, the location and amplitude for all $[Ca^{2+}]_i$ peaks in control and SST-ChR2 islets were quantified and normalized to the highest amplitude peak in the entire run.

**Electrophysiological recordings from delta cells**. Acute pancreas slices from SSTxcre-Ai32 ChR2 eGFP mice were prepared by embedding pancreas tissue in low-melting agarose. Tissue was sliced in ice-cold cutting solution containing (in mM: sucrose 250, $NHCO_3$ 26, glucose 10, $MgCl_2$ 4, myo-inositol 3, KCl 2.5, sodium pyruvate 2, $NaH_2PO_4$ $2H_2O$ 1.25, ascorbic acid 0.5, $CaCl_2$ 0.1, and kynurenic acid 1), then kept at room temperature (23 ℃) in ACSF (in mM: NaCl 126, $NaHCO_3$ 24, $NaH_2PO_4$ 1, KCl 2.5, $CaCl_2$ 2, $MgCl_2$ 2, glucose 4, ascorbic acid 0.4, 300–310 mOsm) bubbled with a 95% $O_2$/5% $CO_2$ mixture. Delta cells were identified by their eYFP expression and their electrical signals were measured by standard whole-cell patch-clamp recordings (10–15 MΩ resistance electrodes) and K-gluconate based internal solution (in mM: K-gluconate 130, KOH 10, $MgCl_2$ 2.5, HEPES 10, $Na_2ATP$ 4, $Na_3GTP$ 0.4, EGTA 5, $Na_2$phosphocreatinin 5, 290–295 mOsm, 7.4 pH). Cells were current clamped at a resting potential of −70 to −75 mV and cell voltage responses to a brief light flash of a mercury arc lamp (465–495 nm, 10 ms, 1–10 Hz) were recorded via a Multiclamp 700B amplifier with Digidata 1440 A interface (Molecular Devices).

**Electron microscopy of mouse and human pancreas**. Small volumes of a mouse and a human pancreas were fixed and prepared for electron microscopy as described in reference[49]. Islet sections were imaged with scanning and transmission electron microscopy and delta cells identified using established guidelines for islet electron micrographs. 3D electron tomograms of mouse delta cell filopodia were acquired as described in reference[49] and their structure was reconstructed using IMOD software (University of Colorado).

**Statistics**. All statistics were performed using Graphpad Prism 7 and unless otherwise noted, testing of significance between experimental groups was done using OneWay Anova with a two-stage linear step-up procedure of Benjamini, Krieger, and Yekutieli post test to control for the false discovery rate (FDR) with a Q value between 0.01 and 0.005.

**Reporting summary**. Further information on research design is available in the Nature Research Reporting Summary linked to this article.

## Data availability
The source data for Fig. 1a,c–e and g, 2d, e, and h, 3d–f, 4e, f, 5b, c and Supplementary Figs 1a and e, 7f, 8a–c, e, f and 9a–d are provided as Source Data file. The raw data that support the findings of this study are available from the corresponding author upon reasonable request.

## Code availability
The codes utilized to uncover/support the findings of this study are available from the corresponding author upon reasonable request.

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

## Acknowledgements

This work was supported by the Lee Kong Chian School of Medicine, Nanyang Technological University start-up grant to Per-Olof Berggren, the Lee Foundation grant, the Swedish Research Council, the Family Erling-Persson Foundation, the Novo Nordisk Foundation, the Stichting af Jochnick Foundation, the Swedish Diabetes Association, the Scandia Insurance Company Ltd., Diabetes Research and Wellness Foundation, Berth von Kantzow's Foundation, the Strategic Research Program in Diabetes at Karolinska Institutet, the ERC-2013-AdG 338936-Betalmage, and the Knut and Alice Wallenberg Foundation. Rafael Arrojo e Drigo was supported by a research fellowship from the Nanyang Structural Biology Institute (NISB) and by an AXA Research Fund postdoctoral fellowship. Flavia Letícia Martins Peçanha was supported by a grant from the National Council for Scientific and Technological Development—Brazil (CNPq). The authors would like to thank Javier Chow from The Salk Institute for assistance with implementing MATLAB scrips for delta cell activity analysis. P-OB is co-founder and CEO of Biocrine, an unlisted biotech company that is using the ACE technique as a research tool, S.J. was employed by Biocrine AB and I.L. is a consultant for Biocrine AB. Open access funding provided by Karolinska Institute.

## Author contributions

RAD designed the study, acquired, analyzed and interpreted the data, and wrote the article. S.J., X.Z., C.F.G-P., M.F., H.T.T.N., O.S., F.L.M.P., R.R-D., E.B., T.D., S.P., Y.A., I.L., M.C., T.B., S-H.S., M.G., G.J.A. and M.H.E. contributed to the acquisition of data and critical revision of the article. P-O.B. was the originator of the idea underlying this manuscript and contributed to the design and writing of the article. All authors approved the final version to be published. P-O.B. is the guarantor of this work.

## Additional information

**Competing interests:** The authors declare no competing interests.

