## [Peer Review File · Nature Communications]

Reviewers' comments:

Reviewer #1 (Remarks to the Author):

Thank you for giving me the opportunity to see this interesting manuscript once again.

This is a resubmission of a paper originally submitted to another journal and that has been transferred to Nature Communications.

The authors have explored the function of the pancreatic delta-cells by a series of elegant and technically challenging microscopy technique, photoactivation ('optogenetics' and calcium imaging).

Overall, this is an impressive study. As with the original submission, I find the most interesting finding to be the dynamic regulation of the filopodia. They have now extended this bit and show that these filopodia are regulated over minutes/hours, that the filopodia extending to the blood vessels are stable but those that connect with other endocrine cells more dynamic.

However, I must say that the condensed (letter to Nature) style makes this manuscript extremely difficult to read and in some places the authors' argument is somewhat difficult to follow. In fact, some of the data presented in the (supplementary) figures are not mentioned (let alone) discussed. I strongly advise the authors - now that they are not confined to the tight word limit - prepare a full manuscript of the data and describe the rationale of the experiments, the underlying hypotheses and findings more extensively.

Specific comments

Page 1. Reference 14. I think we have much better data now on the expression of different SSTRs based on RNAseq of both mouse and human cells. I don't think - as suggested in reference 14 - that there is much difference between alpha- and beta-cells. Why do the authors suspect that 'molecular machinery' should be different in vivo?

Page 2. 'Surrogate for delta-cell activity' - surrogate for what? Secretion, electrical activity?

Page 2: Glucose concentrations. I am afraid that the protocol is extremely confusing. As I understand it, the authors use insulin to lower glucose to 5.5 mM and then inject glucose. It would be much easier to follow if approximate glucose concentrations were inserted in Fig. 1. Supplementary Fig. 1A if anything makes things even more confusing. If glucose was injected at $t=0$, then glucose should not increase until $t>0$ but the graph (erroneously, I think) indicates it increases from $t=-300s$.

Page 2: 'Because these $[Ca^{2+}]_i$ dynamics are different from those of beta-cells in vivo, we conclude that the delta cells are heterogeneous'. How the first and second premise lead to the conclusion is confusing!! (dynamics - like almost all words of Greek origin are singular even when they end with an 's')

Page 3: I think the evidence that Ucn3 is a coupling factor is overstated. In the van Meulen paper, it is clear that the reduction of glucose-induced somatostatin secretion is almost entirely due to a reduction of somatostatin content. When the response to glucose was normalised to content, there was no difference between wildtype and Ucn3 knockout. Thus, Ucn3 is a transcription factor regulating somatostatin expression but it is not THE coupling factor. This is not evident from the way it is phrased here!

Page 4: The description of the 'hierarchical cluster analysis' is also too brief. Although I have used this myself, it is not evident what the authors are trying to show in Fig 1F. Neither the main text/figure legend, nor the methods section provides much guidance.

Fig. 2. The authors make a heroic effort to explore how the delta cells filopodia are regulated but from the data it is really not clear whether it is insulin, Igf1 or Vegfa. The impression one gets

from reading this is that 'we don't know'. Some explanation why the authors decided to investigate these three substances would perhaps help. In the meantime, just some data on whether delta-activity (by photoactivation or just glucose stimulation in vitro) would provide a good start.

What is shown in Fig. 2E. In Supplementary Fig. 9F similar analyses are shown and then said to be 'simulations' - is that the case here too? I tried to find out from the Methods how these simulations were made but found no details. Of course, the finding that the longer the filopodia get, the more contact they are able to make is somewhat trivial but the authors make a strong statement about this 'numerically disadvantaged delta cells etc.)

Fig. 3. The calcium data in Fig. 3 are truly impressive but again it would help if approximate glucose concentrations were given in the graphs. Are these data based just on 5 cells in 4 islets from 1 or several mice?). Although these experiments most likely are technically challenging, the statistical material is perhaps a bit small.

It is very interesting that secretion can occur in the filopodia.

Fig. 4. I think the layout of the figures and the legends are different from what is described in the text and that reference to Fig. 4A is really to Fig. 4C etc.?

Here the use of photoactivation of delta-cells expressing channelrhodopsin to study the impact on beta-cell function (monitored as changes in calcium). The extended data show that the delta-cell responses depend on the stimulation frequency and that there at the frequency goes >5Hz, there is a dramatic reduction in the delta cell responsiveness. This is not commented on in the text and what type of stimulation protocol was used in Fig. 4C (1, 5, 10 Hz or even continuous light activation?) is not stated. The graph contains a break but there is no indication of how long it is - but this is important because the effect of somatostatin on beta-cell electrical activity is very transient.

Fig. 4F-H. The experiments are interesting but it would be nice if they were complemented by measurements of somatostatin release (in vitro).

The calcium data in Fig. 4F are difficult to interpret. It is said to be the 'normalised' responses but it is not made clear what they are normalised to (the maximum amplitude? Just looking at the data (e.g. lower yellow trace in WD) suggest that calcium is actually not that different but just appears so because of a single very large spike. Given this, it is also difficult to evaluate the data in Fig. 4G (fold what?).

Fig. 4H. This is interesting - but is not possible to get data from human type 2 patients? The significance of 'increasing their reach' is not clear if we don't know anything about secretion. Does it at all matter if there is no secretion in the first place (as perhaps suggested by the calcium measurements).

Extended data 5. colocalization. the somatostatin labelling is diffuse - not granular. How certain can we be then that there is a true co-localization of somatostatin with Cav1.2, vamp2 etc and that it is not just out of focus fluorescence?

Legend Figure 1. 1080 minutes - I suspect it is not quite as long as that and that it should be 1080 s

Reviewer #2 (Remarks to the Author):

The revised manuscript added new pieces of data and overall presented a detailed picture of the electrical activities and morphological changes of delta cells under normal and high-fat induced diabetic conditions. These data showed convincingly that delta cells possess cellular protrusions (filopodia) and that these filopodia are dynamic structures. However, it remains unclear whether

morphological changes of filopodia actually impact delta cell function. Establishing such causal relationship may be technically very challenging as it requires manipulating filopodia length and numbers in a controlled manner coupled with functional analysis. The data in this manuscript are therefore largely descriptive in nature but nonetheless provide useful information for the field, in particular, the calcium imaging data with fine resolution and live imaging of delta cell morphological changes are previously not available. Given the lack of structure-function causality here, I suggest the authors re-evaluate some of the language used to describe these data and its implications.

I found Fig. 4 and the text associated with Fig. 4 to be confusing. The data in Fig4. C-D are from beta cells whereas data in Fig.4 F-H are from delta cells. These figures should be labelled clearly with the different islet cells analyzed and I further suggest a diagram (more informative than Fig. 4B) to help the readers understand the experimental schema here. In addition, the text calls for Figure 4 are all wrong, need to be fixed.

The authors used "slow" and "oscillatory" to describe the same type of delta cells, please use just one to avoid confusion.

Reviewer #3 (Remarks to the Author):

The core of the paper is the concept of dynamic delta cell filopodia that are making contact with multiple beta and alpha cells in the islet, and are participating in somatostatin release that in turn controls beta cell function.

This is a fascinating and provocative suggestion, one that may change the way we think about islet biology.

In reading the response of the authors to the comments of the reviewers, in particular reviewer 2, it seems that restructuring the paper has improved it considerably.

It is true that many details in the model are missing and some of the demonstrations of filopodia and their function are suboptimal, but I nonetheless think that on a balance, this is a worthy paper for Nature Communications.

One minor comment- the authors suggest that endogenous insulin, IGF1 and VEGF signaling impact the delta cell filopodia. While insulin and IGF1 could act directly on delta cells, to the best of my knowledge VEGF receptors are not expressed in delta cells, suggesting a non-autonomous pathway leading from beta cells (the most abundant source of VEGF in islets), through activation of VEGF receptors on vascular endothelial cells, to modulation of delta cell morphology (similar conceptually to pathways shown in liver dynamics). The authors may want to discuss this briefly (or alternatively, show that VEGFR1/2 are expressed in delta cells).

Reviewers' comments:

Reviewer #1 (Remarks to the Author):

Thank you for giving me the opportunity to see this interesting manuscript once again.

This is a resubmission of a paper originally submitted to another journal and that has been transferred to Nature Communications.

The authors have explored the function of the pancreatic delta-cells by a series of elegant and technically challenging microscopy technique, photoactivation ('optogenetics' and calcium imaging.

Overall, this is an impressive study. As with the original submission, I find the most interesting finding to be the dynamic regulation of the filopodia. They have now extended this bit and show that these filopodia are regulated over minutes/hours, that the filopodia extending to the blood vessels are stable but those that connect with other endocrine cells more dynamic.

However, I must say that the condensed (letter to Nature) style makes this manuscript extremely difficult to read and in some places the authors' argument is somewhat difficult to follow. In fact, some of the data presented in the (supplementary) figures are not mentioned (let alone) discussed. I strongly advise the authors - now that they are not confined to the tight word limit - prepare a full manuscript of the data and describe the rationale of the experiments, the underlying hypotheses and findings more extensively.

Thank you for your comments and support. As suggested by the reviewer, we have re-written the manuscript.

Specific comments

Page 1. Reference 14. I think we have much better data now on the expression of different SSTRs based on RNAseq of both mouse and human cells. I don't think - as suggested in reference 14 - that there is much difference between alpha- and beta-cells. Why do the authors suspect that 'molecular machinery' should be different in vivo?

Answer: We agree with the reviewer and the initial sentence was a bit confusing. We have revised the text and made the following edit:

(line 52)

Somatostatin is a powerful paracrine inhibitor of islet alpha and beta cells, which express different isoforms of the somatostatin receptor. These receptors enable somatostatin-dependent suppression of glucagon and insulin release(Strowski et al., 2000), indirectly impacting on glucose homeostasis(Hauge-Evans et al., 2009, 2015).

Page 2. 'Surrogate for delta-cell activity' - surrogate for what? Secretion, electrical activity?

Answer: We meant as a surrogate for delta cell activity and somatostatin release. We have updated the text and now it reads:

(line 70)

"(...) as a surrogate of delta cell activity and somatostatin release(...)"

Page 2: Glucose concentrations. I am afraid that the protocol is extremely confusing. As I understand it, the authors use insulin to lower glucose to 5.5 mM and then inject glucose. It would be much easier to follow if approximate glucose concentrations were inserted in Fig. 1. Supplementary Fig. 1A if anything makes things even more confusing. If glucose was injected at $t=0$, then glucose should not increase until $t>0$ but the graph (erroneously, I think) indicates it increases from $t=-300$ s.

Answer: We have revised both figures and made the modifications suggested by the reviewer.

Page 2: 'Because these $[Ca^{2+}]_i$ dynamics are different from those of beta-cells *in vivo*, we conclude that the delta cells are heterogenous'. How the first and second premise lead to the conclusion is confusing!! (dynamics - like almost all words of Greek origin are singular even when they end with an 's')

Answer: We have revised the text and modified this part of the text. Now, our conclusion regarding the two different "types" of delta cell activity reads:

(line 87)

"At first, these slow $[Ca^{2+}]_i$ dynamics may resemble those observed in beta cells *in vivo* (Chen et al., 2016) under similar conditions. However, and differently from beta cells, slow $[Ca^{2+}]_i$ dynamics cells displayed a significant reduction in $[Ca^{2+}]_i$ frequency and amplitude ~3-minutes post-stimulation, akin to spiking delta cells (Fig. 1B-E). Together, these results indicate that there are two different types of delta cells that can be characterized by the profile of their $[Ca^{2+}]_i$ in response to glucose stimulation."

Page 3: I think the evidence that Ucn3 is a coupling factor is overstated. In the van Meulen paper, it is clear that the reduction of glucose-induced somatostatin secretion is almost entirely due to a reduction of somatostatin content. When the response to glucose was normalized to content, there was no difference between wildtype and Ucn3 knockout. Thus, Ucn3 is a transcription factor regulating somatostatin expression but it is not THE coupling factor. This is not evident from the way it is phrased here!

Answer: We have re-written the text and now it reads:

(line 107)

". While the identity of the factors that allow delta cell synchronization remains to be determined, it is possible that factors secreted from neighboring cells, the autonomic nervous system and gap junction communication may contribute to delta cell synchronization (Briant et al., 2018; DiGrucchio et al., 2016; Molina et al., 2014)".

Page 4: The description of the 'hierarchical cluster analysis' is also too brief. Although I have used this myself, it is not evident what the authors are trying to show in Fig 1F. Neither the main text/figure legend, nor the methods section provides much guidance.

Answer: We have revised the text and included a more detailed explanation of the clustering analysis' rationale and interpretation:

(line 95)

"Our results suggest that under basal conditions the activity of individual delta cells within the islet is "uncoordinated" (e.g. each delta cell can fire at different time points) and that glucose

stimulation leads to the synchronization of delta cell $[Ca^{2+}]_i$ spiking activity (Fig.1B and Movie1-2). We tested this hypothesis by determining the degree of correlation between delta cells followed by hierarchical cluster analysis (HCA) during basal ($t=0-2min$), glucose-stimulated ($t=2-6min$) and recovery states ($t>6min$). This approach considers the activity profile of each delta cell during the specified time points and determines the degree of correlation (or similarity) which is then ordered by hierarchical clustering. We observed that glucose increases the number of delta cells with similar activity patterns up to 4 minutes after stimulation, thus creating highly synchronized delta cell clusters within the islet (Fig. 1F).”

Fig. 2. The authors make a heroic effort to explore how the delta cells filopodia are regulated but from the data it is really not clear whether it is insulin, Igf1 or Vegfa. The impression one gets from reading this is that 'we don't know'. Some explanation why the authors decided to investigate these three substances would perhaps help.

Answer: We have revised the text and included the following paragraph to address the rationale for choosing these factors:

(line 165)

“We next considered the chemical signals that could drive filopodia dynamics. We focused on three factors secreted by beta cells, namely insulin, IGF-1(Maake and Reinecke, 1993) or VEGF-A(Brissova et al., 2006). Insulin and IGF-1 were chosen because they are secreted in abundance by beta cells. VEGF-A was chosen since it is a beta-cell secreted factor that acts on endothelial cells, which are in turn targeted by the delta cell filopodia (Fig.2G, Extended Data 4). Importantly, delta cells express insulin and IGF-1 receptors (IGF-1R)(DiGrucchio et al., 2016).”

In the meantime, just some data on whether delta-activity (by photoactivation or just glucose stimulation in vitro) would provide a good start.

Answer: We would like to point the reviewer to previous studies that characterized the activity of delta cells in Sst-ChR2 mice (Lenguito et al., 2017) or subsequent to glucose stimulation (Briant et al., 2018; Zhang et al., 2007).

What is shown in Fig. 2E. In Supplementary Fig. 9F similar analyses are shown and then said to be 'simulations' - is that the case here too? I tried to find out from the Methods how these simulations were made but found no details. Of course, the finding that the longer the filopodia get, the more contact they are able to make is somewhat trivial but the authors make a strong statement about this 'numerically disadvantaged delta cells etc.)

Answer: Yes, the experiments indicated by the reviewer are “simulations” that calculate the number of cells that would be in direct contact with delta cells. We have revised the text and we provide a detailed explanation in the methods sections of how this simulation is done. In addition, we included a graphical illustration to Extended Data 3J that illustrates the simulation process.

The new method section reads:

(line 380)

“Simulation of delta cell contact densities and reach in islet sections. Mouse and human islets immunostained with anti-insulin, -glucagon and/or -somatostatin antibodies and DAPI were mapped using an imageJ plugin as previously described (26). In brief, the nuclei and cells are segmented in 3D, which generates a “geographical” map of alpha, beta and/or delta cells in XY and Z planes of each islet (Extended Data 3A-C). Next, we determined the number of direct

delta cell contacts (characterized by adjacent cell boundaries) with neighboring alpha or beta cells. This information is translated into a measure of “contact frequency”, which is the percentage of alpha or beta cells contacted by a given delta cell (e.g. if a delta cell contacts 2 out of 20 total mapped alpha cells, that delta cell’s contact frequency with alpha cells is 10%). This process is repeated for each delta cell mapped. Next, once the position of each alpha and/or beta and delta cell is known, we simulated the expected contact density that would be observed if the boundaries of each delta cell were expanded by 1µm increments in all possible directions (except towards the outside of the islet structure) until it reaches a maximum “filopodia length” value determined by the user (Extended Data 3J).”

The new figure legend contains the following description:

“(J) Graphical illustration of how the delta cell reach is determined after all alpha, beta and delta cells are mapped. For each delta cell (white, center), we generate radial expansions (pink dotted lines) that increase in diameter with 1µm steps to simulate the gradual increase in the number of cells that would be within reach of the delta cell filopodia. In this cartoon, cell 1 gets within reach at the 2µm mark, while cell 2 is reached at the 4µm mark.”

Fig. 3. The calcium data in Fig. 3 are truly impressive but again it would help if approximate glucose concentrations were given in the graphs. Are these data based just on 5 cells in 4 islets from 1 or several mice?). Although these experiments most likely are technically challenging, the statistical material is perhaps a bit small.

It is very interesting that secretion can occurs in the filopodia.

Answer: We have revised the text and included the following paragraph:

(line179)

“We asked whether delta cell filopodia have a secretory/regulatory role. From our in vivo TPM imaging dataset (Fig.1), we identified a total of 5 delta cells with discernable filipodia in 4 different SST-GCaMP3 islets (n=1 islet/mouse). Under resting conditions, delta cell somata and filopodia displayed low amplitude $[Ca^{2+}]_i$ spikes that were not synchronized and that had generally lower intensity in the filopodia (Fig. 3A).”

Fig. 4. I think the layout of the figures and the legends are different from what is described in the text and that reference to Fig. 4A is really to Fig. 4C etc.?

Answer: Thank you for pointing this out and we apologize for the confusion. We have revised the manuscript and now the correct references/layout is presented.

Here the use photoactivation of delta-cells expressing channelrhodopsin to study the impact on beta-cell function (monitored as changes in calcium). The extended data show that the delta-cell responses depend on the stimulation frequency and that there at the frequency goes >5Hz, there is a dramatic reduction in the delta cell responsiveness. This is not commented on in the text and what type of stimulation protocol was used in Fig. 4C (1, 5, 10 Hz or even continuous light activation?) is not stated. The graph contains a break but there is no indication of how long it is - but this is important because the effect of somatostatin on beta-cell electrical activity is very transient.

Answer: The break in Figure 4B represents a 4x15-second stimulation window using a 450-490 nm light. Of note, this 15Hz frequency was the fastest “on/off” speed allowed by the shutter equipped with the 450/490nm light used for these experiments. This is described in the methods section (line356).

Fig. 4F-H. The experiments are interesting but it would be nice if they were complemented by measurements of somatostatin release (in vitro).

Answer: This point is well taken. However in the actual experiments we were choosing $[Ca^{2+}]_i$ as a highly sensitive integrative signal for pancreatic delta cell function that indeed could be evaluated *in vivo*.

The calcium data in Fig. 4F are difficult to interpret. It is said to be the 'normalized' responses but it is not made clear what they are normalized to (the maximum amplitude? Just looking at the data (e.g. lower yellow trace in WD) suggest that calcium is actually not that different but just appears so because of a single very large spike. Given this, it is also difficult to evaluate the data in Fig. 4G (fold what?).

Answer: The $[Ca^{2+}]_i$ data shown in Fig4F is normalized to the peak with the highest amplitude for each cell. This approach was also used in Fig 1 and it is described in the methods section (lines 451-453) along with the tools and parameters used to extract the *in vivo* Ca^{2+} data.

With regards to the activity of delta cells in WD islets (yellow traces referred to by the reviewer): we would like to point out that, although the activity pattern of WD-delta cells may look similar for most cells regarding their glucose “activation pattern”, the main difference is observed on the amplitude of the Ca^{2+} transients and not in their frequency or even the synchronization between delta cells within the same islet. In fact, most delta cells in WD-islets fail to respond to glucose with an increase in $[Ca^{2+}]_i$ spiking events. In addition, most delta cells from WD islets have significantly lower $[Ca^{2+}]_i$ amplitudes when compared to control cells (Extended Data 8E).

In order to present only the most relevant observation (i.e. impaired glucose-stimulated $[Ca^{2+}]_i$ spiking events) in a concise manner in the main figure (Fig4F), we decided to normalize the observed $[Ca^{2+}]_i$ amplitudes to the average amplitude in the basal (before glucose) condition. Therefore, the data is shown as “fold-change” from the basal state. We have revised the figure legend and it now reads:

(line 570)

“Spiking delta cell Ca^{2+} peak amplitude signal during basal, glucose or recovery time periods normalized to basal conditions in CD (Basal: n=358; Glucose: n=588; Recovery: n=1060 peaks from n=23 cells in 4 islets total) or WD (Basal: n=127; Glucose: n=224; Recovery: n=353 peaks from n=14 cells in 4 islets total) animals. Data shown as fold-change from the basal state.”

Fig. 4H. This is interesting - but is not possible to get data from human type 2 patients? The significance of 'increasing their reach' is not clear if we don't know anything about secretion. Does it at all matter if there is no secretion in the first place (as perhaps suggested by the calcium measurements).

Answer: This is an interesting point raised by the reviewer but at the moment it is unfortunately not possible for us to get data from human type 2 patients.

Extended data 5. colocalization. the somatostatin labelling is diffuse - not granular. How certain can we be then that there is a true co-localization of somatostatin with Cav1.2, vamp2 etc and that it is not just out of focus fluorescence?

Answer: First, we would like to address the “diffused Sst staining” comment. Based on our electron microscopy data, somatostatin granules are tightly packed in the delta cell filopodia (Fig3), sometimes just a few nanometers apart. Therefore, and since we used confocal microscopy (which is limited to ~200nm resolution in X-Y) we are unable to resolve this small space separating most granules, which likely explains the diffuse appearance of Sst staining. Nevertheless, some individual Sst granules - mainly at the edges of the delta cell filopodia - can be seen in human islets (Extended Data 5I).

Second, to address the “true co-localization” comments: all immunohistochemistry data shown in this manuscript was collected using confocal light microscope, which by design suppresses out-of-focus fluorescence during image formation with the pinhole (set to 1 airy unit).

In addition, as described in the methods section “High-resolution confocal imaging of pancreatic islets and imaging processing”, we practiced the following operational standards to collect all co-localization data: i) images were acquired with Nyquist sampling parameters for confocal microscopy, which basically determines the sampling rate required to collect and reconstruct all information from a given signal (in our case, fluorescence) and decrease sampling (<https://svi.nl/NyquistRate>); ii) the sample was mounted in media that closely matched the refractive index of the objective used (63x/1.3 NA, glycerol) and the objective’s correction collar was adjusted prior to each imaging session. These two important steps were implemented to minimize optical aberrations that could interfere with downstream imaging processing methods and our interpretation of the results. Finally, and prior to testing the co-localization of Sst with target proteins, we applied image deconvolution for each fluorescence channel (deconvolution parameters and software utilized are listed in the methods section).

Legend Figure 1. 1080 minutes - I suspect it is not quite a long as that and that it should be 1080s

Answer: The reviewer is correct and we have corrected the figure in question.

Reviewer #2 (Remarks to the Author):

The revised manuscript added new pieces of data and overall presented a detailed picture of the electrical activities and morphological changes of delta cells under normal and high-fat induced diabetic conditions. These data showed convincingly that delta cells possess cellular protrusions (filopodia) and that these filopodia are dynamic structures. However, it remains unclear whether morphological changes of filopodia actually impact delta cell function. Establishing such causal relationship may be technically very challenging as it requires manipulating filopodia length and numbers in a controlled manner coupled with functional analysis. The data in this manuscript are therefore largely descriptive in nature but nonetheless provide useful information for the field, in particular, the calcium imaging data with fine resolution and live imaging of delta cell morphological changes are previously not available. Given the lack of structure-function causality here, I suggest the authors re-evaluate some of the language used to describe these data and its implications.

I found Fig. 4 and the text associated with Fig. 4 to be confusing. The data in Fig4. C-D are from beta cells whereas data in Fig.4 F-H are from delta cells. These figures should be labelled

clearly with the different islet cells analyzed and I further suggest a diagram (more informative than Fig. 4B) to help the readers understand the experimental schema here. In addition, the text calls for Figure 4 are all wrong, need to be fixed.

Answer: We thank the reviewer for pointing this out. We have changed the text associated with figure 4 so all the figure references are now correct.

The authors used "slow" and "oscillatory" to describe the same type of delta cells, please use just one to avoid confusion.

Answer: We have revised the text and adopted the terms "spiking" or "slow" to describe types of delta cells throughout the manuscript.

Reviewer #3 (Remarks to the Author):

The core of the paper is the concept of dynamic delta cell filopodia that are making contact with multiple beta and alpha cells in the islet, and are participating in somatostatin release that in turn controls beta cell function.

This is a fascinating and provocative suggestion, one that may change the way we think about islet biology.

In reading the response of the authors to the comments of the reviewers, in particular reviewer 2, it seems that restructuring the paper has improved it considerably. It is true that many details in the model are missing and some of the demonstrations of filopodia and their function are suboptimal, but I nonetheless think that on a balance, this is a worthy paper for Nature Communications.

One minor comment- the authors suggest that endogenous insulin, IGF1 and VEGF signaling impact the delta cell filopodia. While insulin and IGF1 could act directly on delta cells, to the best of my knowledge VEGF receptors are not expressed in delta cells, suggesting a non-autonomous pathway leading from beta cells (the most abundant source of VEGF in islets), through activation of VEGF receptors on vascular endothelial cells, to modulation of delta cell morphology (similar conceptually to pathways shown in liver dynamics). The authors may want to discuss this briefly (or alternatively, show that VEGFR1/2 are expressed in delta cells).

Answer: This is a similar comment to the one raised by reviewer #1. We have revised the text and included the following text:

(line 165)

"We next considered the chemical signals that could drive filopodia dynamics. We focused on three factors secreted by beta cells, namely insulin, IGF-1(Maake and Reinecke, 1993) or VEGF-A(Brissova et al., 2006). Insulin and IGF-1 were chosen because they are secreted in abundance by beta cells. VEGF-A was chosen since it is a beta-cell secreted factor that acts on endothelial cells, which are in turn targeted by the delta cell filopodia (Fig.2G, Extended Data 4). Importantly, delta cells express insulin and IGF-1 receptors (IGF-1R)(DiGrucchio et al., 2016)."

(line 285)

"Islet endocrine cells have been reported not to express VEGFRs(DiGrucchio et al., 2016). Hence, it is likely that the filopodia phenotype observed after inhibition of VEGF signaling

pathways is mediated indirectly by islet endothelial cells, which remain in isolated islets for days(Nyqvist et al., 2011).

Reviewers' comments:

Reviewer #1 (Remarks to the Author):

Rewriting the manuscript greatly improved readability.

A few minor things require attention:

1) I think the reference to the 2000 Strowski paper is obsolete. We have much better RNAseq data now (see PMID: 27408771 and PMID:27390011)

2) Is the new section on linwa 88-92 really necessary (i.e. up to the sentence starting 'Together...'). It is clear that calcium oscillatory activity is very different in delta cells than in beta-cells (similar to what was recently reported PMID: 30635569)

3) Delta-cell activity is vague (line 70) – electrical or secretory activity meant? Or both?

4) Is it not slightly contradictory that insulin both tended to decrease filopodia length (line 173) and increase 'normal filopodia length in PPP-treated islets'? Would it not have been possible to test high glucose as such?

5) Not always clear how many animals were used; i.e. line 186 - why not say 5 islets from 5 different mice? See also line 593: 'n=14 cells in 4 islets total) animls' Unable to make sense of this sentence!

6) Photoactivation experiments (lines 226-). The experiments using photoactivation of channelrhodopsin are not described in sufficiently great detail. Methods (lines 465-) should include more experimental detail (standard or perforated patch whole-cell), media composition, temperature. I would not refer to the voltage responses as 'action potentials' – they seem just to be the passive response to the current injection (the current should be displayed on the same time base as the voltage changes. There is no comment about the different stimulation frequencies in the text. Transient inhibitory effects of somatostatin on beta-cell electrical activity have also been reported in PMID:8816714.

Line 252: Better to say high-fat diet than 'Western diet'. Effects of long-term exposure to NEFA have been reported (PMID:18622593). Remain unpersuaded that normalising to peak calcium is the best way of presenting the data in Fig. 4F-G. Why not normalise to basal?

Line 303- Discussion; Is it possible from the authors data to explain why somatostatinergic tone appears increased in diabetic islets? (PMID: 22106159). Perhaps add a coment!

Another round of proofreading would not harm! (paying particular attention to usage of lower and upper case and sub/superscripts). Note that it is Student's t-test - not student's!

Reviewer #2 (Remarks to the Author):

As I pointed out in the last review, Fig. 4A-G were not referenced correctly in the text, and this continues to be the case in this revised manuscript.

Fig.4 is an amalgamation of two very different experiments. I suggest separating it into two figures. Again, as I suggested in the last review, it will be very helpful to the readers to have a simple diagram (for example, a delta and a beta cell side-by-side, one with Chr2, one with GCaMP) to illustrate how the experiment in Figure 4A-E was done.

Reviewer #3 (Remarks to the Author):

The authors have addressed my remaining concerns and I support now publication of the paper.
Congratulations for performing a provocative and interesting study!

Reviewers' comments:

Reviewer #1 (Remarks to the Author):

Rewriting the manuscript greatly improved readability.

A few minor things require attention:

1) I think the reference to the 2000 Strowski paper is obsolete. We have much better RNAseq data now (see PMID: 27408771 and PMID:27390011)

Answer: The Strowski et al manuscript is cited here because it contains experimental data gathered from *in vitro* experiments using rodent islets showing the influence of SSTRs in alpha and beta cell secretion. Therefore, and although this reference is almost 20 years old, it is still the best reference to substantiate the claim that SSTRs “enable somatostatin-dependent suppression of glucagon and insulin release”.

We have taken the reviewer’s advice and cited the two recent papers on RNAseq data to substantiate the claim that “islet alpha and beta cells, which express different isoforms of the somatostatin receptor” on lines 52-53 of the introduction.

2) Is the new section on line 88-92 really necessary (i.e. up to the sentence starting ‘Together....’). It is clear that calcium oscillatory activity is very different in delta cells than in beta-cells (similar to what was recently reported PMID: 30635569)

Answer: We have revised the text and removed the new section mentioned by the reviewer and made it clearer:

Line 86 - The remaining ~6% of delta cells exhibited slower, glucose-responsive $[Ca^{2+}]_i$ dynamics (Fig.1B, Extended Data 1D). Cells with slow $[Ca^{2+}]_i$ dynamics displayed a significant reduction in $[Ca^{2+}]_i$ frequency and amplitude ~3-minutes post-stimulation, akin to spiking delta cells (Fig.1B-E). Together, these results indicate that there are two different types of delta cells that can be characterized by the profile of their $[Ca^{2+}]_i$ in response to glucose stimulation.

We also included the recent paper by Vergari et al in the discussion:

Line 281: Besides glucose^{17,18}, activation of delta cells can also be mediated by insulin secreted from neighboring beta cells in response to rising glucose levels⁴⁴.

3) Delta-cell activity is vague (line 70) – electrical or secretory activity meant? Or both?

Answer: Secretory. This distinction has been added to the text.

4) Is it not slightly contradictory that insulin both tended to decrease filopodia length (line 173) and increase ‘normal filopodia length in PPP-treated islets’? Would it not have been possible to

test high glucose as such?

Answer: This point is well taken but it is likely that the insulin receptor gets desensitized after 24 hours of exposure to high insulin and thereby its effects on filopodia compromised. In this context it is not contradictory that insulin restores normal filopodia length in PPP-treated islets. The reason for why high glucose, although stimulating the delta cells, will not do the job is that we need a factor that can change filopodia dynamics and also guide the filopodia to its target cells within the islet. This has now been clarified in the manuscript.

5) Not always clear how many animals were used; i.e. line 186 - why not say 5 islets from 5 different mice? See also line 593: 'n=14 cells in 4 islets total' animals' Unable to make sense of this sentence!

Answer: Thank you for your suggestion. We believe that the reviewer is referring to lines 180 and 577, respectively. We have revised the text and now it reads:

Line 198: "Secretory capacity and $[Ca^{2+}]_i$ dynamics of delta cell filopodia. We next asked whether delta cell filopodia have a secretory/regulatory role. From our in vivo TPM imaging dataset (Fig.1), we identified a total of 5 delta cells with discernable filipodia in 4 different SST-GCaMP3 islets (n=1 islet/mouse, total of 4 different mice)."

Line 662: "peaks from n=14 cells in 4 islets total and 1 islet per animal for a total of 4 animals."

6) Photoactivation experiments (lines 226-). The experiments using photoactivation of channelrhodopsin are not described in sufficiently great detail. Methods (lines 465-) should include more experimental detail (standard or perforated patch whole-cell), media composition, temperature. I would not refer to the voltage responses as 'action potentials' – they seem just to be the passive response to the current injection (the current should be displayed on the same time base as the voltage changes. There is no comment about the different stimulation frequencies in the text. Transient inhibitory effects of somatostatin on beta-cell electrical activity have also been reported in PMID:8816714.

Answer: We followed the reviewer's suggestion and have added additional information about the temperature, media composition and patch-clamp technique ~~during~~ used for our delta-cell recordings (lines 471-477).

The voltage responses that we show are the sum of photocurrents, filtered by the membrane passive properties of the delta cell, as well as action potentials. We are ~~100%~~ confident that action potentials are part of the response, because this component: (1) was overshooting 0 mV; (2) was all-or-none; and (3) had a waveform that closely matched that of action potentials evoked by current pulses; see traces below.

It is inappropriate for us to show the voltage and current responses on the same time scale because they were evoked by light flashes of different durations.

Line 252: Better to say high-fat diet than ‘Western diet’. Effects of long-term exposure to NEFA have been reported (PMID:18622593). Remain unpersuaded that normalizing to peak calcium is the best way of presenting the data in Fig. 4F-G. Why not normalize to basal?

Answer: Thank you for pointing that out. We have revised the text and now we use the term high fat diet instead of western diet, and included a reference to the paper mentioned by the reviewer:

Line 239: “...impaired in islets exposed *in vitro* to free fatty acids³⁸ and in animal models of pre-diabetes and diabetes^{39,40},”

Regarding $[Ca^{2+}]_i$ peak intensities, we chose this normalization approach so the presentation of all the *in vivo* delta cell data throughout the manuscript is consistent and follows the same rationale. This way, one can easily compare the relative amplitude of the $[Ca^{2+}]_i$ peaks in delta cells in control and high-fat diet mice or with the data shown in Figure 1.

Line 303- Discussion; Is it possible from the authors data to explain why somatostatinergic tone appears increased in diabetic islets? (PMID: 22106159). Perhaps add a comment!

Answer: Under pre-diabetic conditions there is an increased basal beta cell secretory activity and hence increased intra-islet insulin levels. Since insulin directly stimulates the delta cell secretory activity this should lead to an increased somatostatinergic tone. This has now been clarified in the manuscript.

Another round of proofreading would not harm! (paying particular attention to usage of lower and upper case and sub/superscripts). Note that it is Student's t-test - not student's!

Answer: We have revised the text one more time and corrected the typos found.

Reviewer #2 (Remarks to the Author):

As I pointed out in the last review, Fig. 4A-G were not referenced correctly in the text, and this continues to be the case in this revised manuscript.

Answer: We apologize for this mistake. We have revised the text and corrected the references to Figure 4.

Fig.4 is an amalgamation of two very different experiments. I suggest separating it into two figures. Again, as I suggested in the last review, it will be very helpful to the readers to have a simple diagram (for example, a delta and a beta cell side-by-side, one with ChR2, one with GCaMP) to illustrate how the experiment in Figure 4A-E was done.

Answer: Thank you for the suggestion. We have split Figure 4 into two figures. Optogenetics experiments are listed in figure 4 (with a diagram describing the experimental setup) and high-fat data is shown in Figure 5.

Reviewer #3 (Remarks to the Author):

The authors have addressed my remaining concerns and I support now publication of the paper. Congratulations for performing a provocative and interesting study!

Answer: Thanks for the support!

Reviewers' comments:

Reviewer #1 (Remarks to the Author):

I am (strongly) supportive of publication of this study. It represents an impressive effort but I would urge to authors to consider the following MINOR points:

1) Line 80, 93, 184 and elsewhere: whenever the word 'basal' is used it help to specify the conditions (i.e. state the approximate glucose concentrations).

Likewise, it should be clarified that the statement (line 84) that 'glucose promoted' means 'increasing glucose to $\sim x$ mM'.

2) Line 140: 'somewhat limited' - isn't this the same as 'modest'. Also, consider removing the sentence on lines 142-145: it is just waffle (sorry!).

3) Line 166: beta-cell or beta cell (hyphen or not - there might be other places - please check for consistency)

4) Lines 225-. They might well be action potentials - it is just that the evidence quoted is not sufficient. The data shown in the rebuttal letter do not address this - here they use current injection, which is not the same as photoactivation!

Why it should be 'inappropriate' to show the voltage responses on an expanded time base is not clear as this would help to address my query. In addition, the zero-voltage and -current levels should be inserted in Fig. 4A. In fact, the amplitude of the voltage responses (80 mV) are precisely that expected for a current of 270 pA and an input resistance of 3GOhm (measured in delta-cells exposed to 1 mM glucose; PMID: 30635569).

Does photoactivation never activate more than one action potential? The responses in Supplementary Fig. 7G could be shown on an expanded time base to address this.

I also note that the electrophysiology was done in the standard whole-cell configuration and at room temperature. I am NOT asking the authors to make additional experiments but I feel it might be - in the absence of better data - prudent to downtone this section.

Very minor: 10-15 MOhm electrodes would not be regarded as high-resistance by most electrophysiologists (this usually refers to resistances of 100-250 MOhm).

There is still no mention of the different photoactivation frequencies (i think this is the 3rd time I request a brief statement to this effect). In fact, the slow deactivation of the light-induced voltage responses as well as the decrease in the amplitude during high frequency stimulation provide good evidence that they are regenerative (action potentials)!

5) Lines 268-272. This section is slightly nebulous. On the one hand there are more filopodia - on the other hand delta-cell activity is reduced. So what is the impact on islet function in terms of insulin/glucagon release? Again - no extra experiments requested - just refrain from making strong statements!

6) The title would have been great for a News-and-views piece but perhaps not so great for an article as it is not very informative (but obviously it is the authors' prerogative).

7) References 23 and 43 look strange. The list of references is a unorthodox in that some references are given with volume and page numbers and other references just contain the DOI. Reference 47 should be updated - the paper was published in 2017!

Reviewer #2 (Remarks to the Author):

This manuscript is much improved after the revisions. I have no further comments.

Reviewer 1:

I am (strongly) supportive of publication of this study. It represents an impressive effort but I would urge to authors to consider the following MINOR points:

Thank you for your support.

1) Line 80, 93, 184 and elsewhere: whenever the word 'basal' is used it help to specify the conditions (i.e. state the approximate glucose concentrations).

Likewise, it should be clarified that the statement (line 84) that 'glucose promoted' means 'increasing glucose to ~x mM'.

Answer: We have revised the text and now the first text reference of “basal” or “glucose-stimulated” conditions specifies the approximate glucose concentration:

Example: Lines 81-82: “Under basal conditions (i.e. 5.5mM glucose)..”

Lines: 84-85: “In these cells, an increase in circulating glucose concentration (from ~5.5mM to ~18mM)”

2) Line 140: 'somewhat limited' - isn't this the same as 'modest'. Also, consider removing the sentence on lines 142-145: it is just waffle (sorry!).

Answer: We have revised the text and now it reads:

Line 140: “While this effect was modest in rodents..”

Line 142-145: These data confirm that the filopodia is a potential avenue of communication between delta cells and other neighboring endocrine cells.

3) Line 166: beta-cell or beta cell (hyphen or not - there might be other places - please check for consistency)

Answer: we have revised the text and removed the hyphenated version of “beta cell”.

4) Lines 225-. They might well be action potentials - it is just that the evidence quoted is not sufficient. The data shown in the rebuttal letter do not address this - here they use current injection, which is not the same as photoactivation!

Why it should be 'inappropriate' to show the voltage responses on an expanded time base is not clear as this would help to address my query.

Answer: To satisfy the curiosity of the reviewer, here is comparison of action potentials evoked by light (blue) and current (black) evoked in the same delta cell. It should be apparent that these two signals are quite similar in their waveforms and, thus, actually are action potentials.

In addition, the zero-voltage and -current levels should be inserted in Fig. 4A.

Answer: For the current trace (Fig. 4A, upper), the zero current level, by convention, is indicated by the baseline prior to the light flash. For the voltage trace (Fig. 4A, lower), the zero voltage level is already indicated by the dashed line; the same is true for the voltage responses shown in Extended Data 7G.

We have revised the text in the legend from both figures in question and included the following line: "Dashed line = 0 mV".

In fact, the amplitude of the voltage responses (80 mV) are precisely that expected for a current of 270 pA and an input resistance of 3GOhm (measured in delta-cells exposed to 1 mM glucose; PMID: 30635569).

Answer: The reviewer should be well aware that ionic currents, and the voltage responses that they produce, depend upon the electrochemical driving force, which is the difference between the membrane potential (V_m) and the reversal potential of the current (E_{rev}):

$$I = g(V_m - E_{rev})$$

Therefore, the Ohmic calculation presented by the reviewer does not apply to membrane potential changes associated with ionic currents.

For ChR2, the reversal potential is approximately 0 mV under physiological conditions (e.g. Nagel et al., 2003; PNAS 100: 13940). During a light-evoked voltage response, the driving force will decrease steadily as the membrane potential depolarizes; at 0 mV (not to mention potentials more positive than 0 mV), there will be no inward current flowing through ChR2 channels and thus no depolarizing drive to the membrane potential. Therefore, an overshooting voltage response, such as the one shown in the lower trace of Fig.4A, cannot be due to the light-activated channels - which would bring the membrane potential to 0 mV but not beyond - and instead must arise from the voltage-activated Na^+ channels that generate action potentials.

Does photoactivation never activate more than one action potential? The responses in Supplementary Fig. 7G could be shown on an expanded time base to address this.

Answer: Prolonged light flashes only evoked single action potentials. To evoke trains of action potentials, it was necessary to use trains of brief light flashes. Expanded traces of action potential responses are shown above.

I also note that the electrophysiology was done in the standard whole-cell configuration and at room temperature. I am NOT asking the authors to make additional experiments but I feel it might be - in the absence of better data - prudent to downtone this section.

Answer: It seems essential for our study to demonstrate that the ChR2 is capable of photostimulating delta cells and this is the purpose of Fig. 4A. As things stand, the presentation of this central point is quite minimal; the entire text related to ChR2-mediated photostimulation of delta cells consists of 2 modest sentences:

"Illumination of ChR2-expressing delta cells with a 470 nm light beam evoked an inward photocurrent (Fig 4A; peak amplitude 273 ± 48 pA at -70 mV; $n = 12$). These currents

depolarized delta cells sufficiently to evoke overshooting action potentials (Fig 4A) and could evoke trains of action potentials at light flash frequencies up to 5 Hz (Extended Data 7G), and maximum somatostatin release³⁴.

It is difficult to imagine how to “downtone” things further.

Very minor: 10-15 MOhm electrodes would not be regarded as high-resistance by most electrophysiologists (this usually refers to resistances of 100-250 MOhm).

Answer: we agree that this is very minor but have revised the text to read:

Line 466: “...electrical signals were measured by standard whole-cell patch-clamp recordings (10-15 MΩ resistance electrodes)...”

There is still no mention of the different photoactivation frequencies (i think this is the 3rd time I request a brief statement to this effect). In fact, the slow deactivation of the light-induced voltage responses as well as the decrease in the amplitude during high frequency stimulation provide good evidence that they are regenerative (action potentials)!

Answer: We have revised the text as indicated:

Line 220: “These currents depolarized delta cells sufficiently to evoke overshooting action potentials (Fig 4A) and could evoke trains of action potentials at light flash frequencies up to 5 Hz (Extended Data 7G), as well as maximum somatostatin release³⁴.”

5) Lines 268-272. This section is slightly nebulous. On the one hand there are more filopodia - on the other hand delta-cell activity is reduced. So what is the impact on islet function in terms of insulin/glucagon release? Again - no extra experiments requested - just refrain from making strong statements!

Answer: We have revised this section and now it reads:

Line 265: “These results indicate that delta cell function is impaired during the onset of hyperglycemia in pre-diabetic animals, a physiological condition associated with beta cell expansion and [Ca²⁺]_i dysfunction. Of note, the longer filopodia observed in pre-diabetic delta cells may reflect adaptation to maintain contact with beta cells in pre-diabetic mice (Fig 5C, Extended Data 8C-F)”.

6) The title would have been great for a News-and-views piece but perhaps not so great for an article as it is not very informative (but obviously it is the authors' prerogative).

Answer: Thank you, but we think that the declarative title is appropriate for this manuscript.

7) References 23 and 43 look strange. The list of references is a unorthodox in that some references are given with volume and page numbers and other references just contain the DOI. Reference 47 should be updated - the paper was published in 2017!

Answer: We apologize for this misstep. The reference list has been updated.

REVIEWERS' COMMENTS:

Reviewer #1 (Remarks to the Author):

In general I am happy with the authors changes but I am slightly surprised by their unwillingness to amend the figures as I think the description of their photoactivation experiments could be enhanced.

I am certainly grateful that the authors 'satisfy [my] curiosity' but it is just possible that other readers may share my curiosity and they will not have the advantage of the rebuttal letter. I think the action potential measurements should be included in the paper (or at least as a supplementary figure).

I debate the authors' statement that the fact that the voltage responses overshoot as such constitutes evidence that they reflect action potentials - this will depend on the reversal potential of the ChR2 current is zero mV (which it may not be! see Fig. 3a Annu. Rev. Biophys. 2015.44:167-186). However, I accept that the light-activated changes in membrane potential are action potential-like.

However, I maintain that this entire section should be downtoned/revise (it did not just refer to the 2-3 lines as the authors seem to think!). This is for two reasons:

1) The electrophysiological measurements shown are conducted using the standard whole-cell technique after hyperpolarising the delta-cells to -70 mV (or more). The impact of photostimulation in the intact cell when the cell is spontaneously active is not known.

2) The authors state that each 15-s flash evokes only a single action potential. If this were true, it would explain the very weak effects on beta-cell function they observe. As the authors lecture me in their rebuttal, the membrane potential will be clamped at E_{Rev} during photo activation, which may be as positive as +10 mV (probably accounting for the stimulation of somatostatin secretion in Ref. 34). However, this has nothing to do with action potential firing. If they insist on this, they must conduct additional experiment adjusting the light intensity such that the depolarisation evokes action potential firing (I would guess that a few pA of current is sufficient; 1% of the currents activated by the protocol they used now!

The authors also (wrongly, in my opinion) state that there is a 'convention that the zero current is the baseline'. Probably the leak current is very small with the conditions used so no remedial action needed.

There remains problems with the proofreading of the manuscript.

Legend Fig. 4. Text in (A) and (D) seem to refer to the same data. The E-G in the legend refers to data in panes D-F

Line 404: KCL should be KCl: lines 490-2: Subscripts in chemical formula missing.

REVIEWERS' COMMENTS:

Reviewer #1 (Remarks to the Author):

In general I am happy with the authors changes but I am slightly surprised by their unwillingness to amend the figures as I think the description of their photoactivation experiments could be enhanced.

I am certainly grateful that the authors 'satisfy [my] curiosity' but it is just possible that other readers may share my curiosity and they will not have the advantage of the rebuttal letter. I think the action potential measurements should be included in the paper (or at least as a supplementary figure).

As requested by the reviewer, we have added a new figure (Extended Data 7G) to compare the waveforms of action potentials evoked by light flashes and by current pulses. The striking similarity of these two waveforms conclusively demonstrates that light evokes action potentials in ChR2-expressing delta cells, a seemingly obvious point that the reviewer has finally accepted.

I debate the authors' statement that the fact that the voltage responses overshoot as such constitutes evidence that they reflect action potentials - this will depend on the reversal potential of the ChR2 current is zero mV (which it may not be! see Fig. 3a Annu. Rev. Biophys. 2015.44:167-186). However, I accept that the light-activated changes in membrane potential are action potential-like.

We thank the reviewer for finally agreeing with what we have been accurately (and non-controversially) stating all along.

However, I maintain that this entire section should be downtoned/revised (it did not just refer to the 2-3 lines as the authors seem to think!).

We remain baffled by the reviewer's concerns; there is literally no mention of action potentials at any point in the ms. beyond the sentences describing light-evoked action potentials (lines 220-226).

This is for two reasons:

1) The electrophysiological measurements shown are conducted using the standard whole-cell technique after hyperpolarising the delta-cells to -70 mV (or more). The impact of photostimulation in the intact cell when the cell is spontaneously active is not

known.

2) The authors state that each 15-s flash evokes only a single action potential. If this were true, it would explain the very weak effects on beta-cell function they observe. As the authors lecture me in their rebuttal, the membrane potential will be clamped at E_{Rev} during photo activation, which may be as positive as +10 mV (probably accounting for the stimulation of somatostatin secretion in Ref. 34). However, this has nothing to do with action potential firing. If they insist on this, they must conduct additional experiment adjusting the light intensity such that the depolarisation evokes action potential firing (I would guess that a few pA of current is sufficient; 1% of the currents activated by the protocol they used now!

Taking together points (1) and (2), we surmise that the reviewer is concerned about the relevance of the *in vitro* recordings shown in Fig. 4A to the *in vivo* results shown in Figs. 4D-F. We feel that this is an unnecessary concern because there is no *a priori* reason to suppose that light will not evoke action potentials in ChR2-expressing delta cells *in vivo*. To acknowledge this issue, and “downtone” the description of action potential recordings, we have added the qualifier “In pancreas slices” to the beginning of the sentence (line 220) describing the *in vitro* action potential recordings.

The authors also (wrongly, in my opinion) state that there is a 'convention that the zero current is the baseline'. Probably the leak current is very small with the conditions used so no remedial action needed.

OK; leak current was indeed small, as is typically the case when recording near the resting potential.

There remains problems with the proofreading of the manuscript.

Legend Fig. 4. Text in (A) and (D) seem to refer to the same data. The E-G in the legend refers to data in panes D-F

Line 404: KCL should be KCl; lines 490-2: Subscripts in chemical formula missing.

We thank the reviewer for catching these minor typos, which have been corrected.